# Pathogen-specific innate immune response patterns are distinctly affected by genetic diversity

Antje Häder[1,12], Sascha Schäuble [2,12], Jan Gehlen [3], Nadja Thielemann [4], Benedikt C. Buerfent [3,5], Vitalia Schüller[5], Timo Hess[3,5], Thomas Wolf[6], Julia Schröder [5], Michael Weber[1,6,7], Kerstin Hünniger[1,4], Jürgen Löffler[8], Slavena Vylkova [9], Gianni Panagiotou [2,10,11], Johannes Schumacher [3,5,13] ✉ & Oliver Kurzai [1,4,13] ✉

Innate immune responses vary by pathogen and host genetics. We analyze quantitative trait loci (eQTLs) and transcriptomes of monocytes from 215 individuals stimulated by fungal, Gram-negative or Gram-positive bacterial pathogens. We identify conserved monocyte responses to bacterial pathogens and a distinct antifungal response. These include 745 response eQTLs (reQTLs) and corresponding genes with pathogen-specific effects, which we find first in samples of male donors and subsequently confirm for selected reQTLs in females. reQTLs affect predominantly upregulated genes that regulate immune response via e.g., NOD-like, C-type lectin, Toll-like and complement receptor-signaling pathways. Hence, reQTLs provide a functional explanation for individual differences in innate response patterns. Our identified reQTLs are also associated with cancer, autoimmunity, inflammatory and infectious diseases as shown by external genome-wide association studies. Thus, reQTLs help to explain interindividual variation in immune response to infection and provide candidate genes for variants associated with a range of diseases.

Human monocytes are integral to the innate immune response and pivotal in first-line defenses such as phagocytosis, cytokine and chemokine release, and antigen presentation[1]. Activation of monocytes is triggered by pattern recognition receptors (PRRs) that sense conserved pathogen-associated molecular patterns (PAMPs). Monocytes express multiple PRRs and are activated by many stimuli. Although microbial recognition is redundant, some PAMPs are restricted to defined pathogen groups such as Gram-negative bacterial lipopolysaccharide (LPS), Gram-positive bacterial lipoteichoic acid, or fungal 1,3-β-D-glucan. In addition to pathogen-dependent variation, the innate

[1]Research Group Fungal Septomics, Leibniz Institute for Natural Product Research and Infection Biology-Hans Knoell Institute, 07745 Jena, Germany. [2]Department of Microbiome Dynamics, Leibniz Institute for Natural Product Research and Infection Biology-Hans Knoell Institute, 07745 Jena, Germany. [3]Institute of Human Genetics, Philipps University of Marburg, 35033 Marburg, Germany. [4]Institute for Hygiene and Microbiology, Julius Maximilians University of Wuerzburg, 97080 Wuerzburg, Germany. [5]Institute of Human Genetics, University of Bonn, School of Medicine & University Hospital Bonn, 53127 Bonn, Germany. [6]Systems Biology and Bioinformatics Unit, Leibniz Institute for Natural Product Research and Infection Biology-Hans Knoell Institute, 07745 Jena, Germany. [7]Institute of Molecular Pathogenesis, Friedrich-Loeffler-Institute, 07743 Jena, Germany. [8]Department of Internal Medicine II, University Hospital Wuerzburg, Josef-Schneider-Strasse 2 /C11, 97080 Wuerzburg, Germany. [9]Research Group Host Fungal Interfaces, Septomics Research Center and Leibniz Institute for Natural Product Research and Infection Biology-Hans Knoell Institute, 07745 Jena, Germany. [10]Faculty of Biological Sciences, Friedrich Schiller University, 07743 Jena, Germany. [11]Department of Medicine and State Key Laboratory of Pharmaceutical Biotechnology, University of Hong Kong, Hong Kong SAR, China. [12]These authors contributed equally: Antje Häder, Sascha Schäuble. [13]These authors jointly supervised this work: Johannes Schumacher, Oliver Kurzai. ✉e-mail: johannes.schumacher@uni-marburg.de; okurzai@hygiene.uni-wuerzburg.de

immune response varies among individuals due to genetic diversity. For example, cytokine production capacity or cell-surface expression of markers used to identify leukocyte differentiation and activation are modulated by genetic variation[2,3].

The impact of genetic variation can be identified by analyzing expression quantitative trait loci (eQTL), or alleles of genetic markers that correlate with gene expression levels. eQTL effects are context-specific and differ with exposure to endogenous and exogenous pathogens[4–6]. Response eQTLs (reQTLs) have been identified that change expression only with pathogen exposure and are often cell-type-specific. With regard to peripheral blood immune cells, reQTLs in monocytes have been shown to outnumber those in other peripheral blood mononuclear cells[7]. reQTLs characterize how genetic host variability influences immune responses and help to functionally characterize risk variants for complex diseases identified in genome-wide association studies (GWAS)[4,5,8]. Notably, most risk variants are in noncoding regions and have gene-regulatory effects[9].

Several reQTL studies investigated the immune response to purified microbial ligands that induce receptor-specific signaling cascades, most prominently LPS to stimulate toll-like receptor 4 (TLR4) signaling[4,5,8]. These studies identified reQTLs, corresponding genes, networks, and signaling pathways downstream of LPS stimulation. Although these expression and eQTL studies deepened our understanding of the genetic regulation of innate immune responses, they do not reflect the complex immune response to whole pathogens. Moreover, little is known about how and to what extent genetic host variability influences the transcriptional immune response to different pathogens, especially the differences to fungal *versus* bacterial pathogens[10,11].

In this work, we study response patterns of primary human blood monocytes from 215 male donors to characterize genetic regulation of innate immune responses to diverse pathogens. We investigate the activation after exposure to a fungal (*Aspergillus fumigatus*) and two bacterial pathogens (Gram-negative *Neisseria meningitidis*, Gram-positive *Staphylococcus aureus*) at the transcriptome and gene-regulatory level. Our results indicate differences between fungal- and bacterial-induced activation of monocytes and, to a lesser extent, between Gram-positive and Gram-negative bacteria. We identify common and pathogen-unique reQTLs and a set of reQTL-regulated genes that are directly related to PRR signaling. Of these we analyze a selected number in females to demonstrate their sex-independent effect. Based on our data on functional relationships between genetic markers and gene expression, we screen reQTLs and corresponding genes to functionally characterize risk variants for complex diseases previously identified by GWAS and describe new links between single-nucleotide polymorphisms (SNPs) and gene expression.

## Results

### Monocyte transcriptome responses discriminate fungal and bacterial pathogens

We first analyzed transcriptome-wide gene expression of monocytes after exposure to *A. fumigatus* (germ tubes), Gram-negative *N. meningitidis* and Gram-positive *S. aureus* to dissect responses to different classes of infectious pathogens and identify pathogen-specific expression patterns. Transcriptomes from monocytes of 215 donors were profiled after 3 h and 6 h pathogen stimulation using 3'-mRNA-sequencing (total 1720 transcriptomes, Supplementary Fig. 1). At this stage we used only male donors to reduce background variation and thus maximize chances to identify significant expression signals, which we can link to genetic variation and subsequently test in a separate female cohort.

The expression profile of the *A. fumigatus*-induced response differed from bacteria-induced gene expression patterns. Despite their distinct cell wall composition, the bacterial pathogens induced similar responses (Fig. 1a). 5505 (after 3 h) and 6616 (after 6 h) differentially

expressed genes (DEGs) showed significant expression differences in stimulated compared to unstimulated monocytes. We observed a considerable pathogen-independent core immune response of 1874 DEGs after 3 h and 2328 DEGs after 6 h of stimulation (Fig. 1b). The response comprised well-known inflammatory markers such as *TNF*, *IL6*, *IFNG*, *PTX3*, and *ICAM1* (Supplementary Data 1). Enrichment analysis of this core immune response using KEGG revealed an over-representation of genes associated with immune receptor pathways including cytokine-cytokine receptor interaction, C-type lectin, and NOD-like receptor signaling pathways, and in signal transduction pathways including TNF, NF-κB, and MAPK signaling pathways (Fig. 1c).

Both bacteria triggered a large common transcriptome response, with 63% (3 h) and 71% (6 h) overlapping DEGs, respectively (Fig. 1b). Within this overlap, enrichment was seen for the JAK-STAT signaling pathway at both timepoints, the RIG-I-like receptor signaling pathway and cytosolic DNA-sensing pathway after 3 h, and Th17 cell differentiation after 6 h. We also identified pathways induced specifically following exposure to *N. meningitidis* (e.g., lysosome-related genes, after 6 h) or *S. aureus* (e.g., glycosaminoglycan biosynthesis, after 3 h) (Fig. 1c).

Compared to the bacterial pathogens, *A. fumigatus* induced a specific immune response divergent to the investigated bacteria. The response was characterized by enrichment of HIF1α (after 3 h and 6 h) and FoxO signaling pathways (after 6 h), cellular processes such as focal adhesion, and essential metabolic pathways including tryptophan and propanoate metabolism (Fig. 1c). In addition, the specific adaptation of monocytes to stimulation with *A. fumigatus* was shown by a metabolic shift to aerobic glycolysis and indicates a glycolytic switch associated with an upregulated glycolysis, leading to production of lactate from glucose even under normoxic conditions (Fig. 1c)[12,13]. Interestingly, this effect was also described in response to alive *A. fumigatus* in macrophages[14], and thus seems to be independent on fungal viability. The glycolytic shift was exemplified by a substantial upregulation in expression of genes encoding glycolytic enzymes in response to *A. fumigatus* after 6 h. High expression of genes encoding lactate dehydrogenases (*LDHA*, *LDHC*, *LDHAL6B*) indicated an increased demand in lactate formation. We also observed increased expression of glucose transporter genes: the strongest for *SLC2A1* (*GLUT1*), encoding glucose transporter 1, after 6 h exposure to *A. fumigatus* (Fig. 1d). *A. fumigatus* also induced higher expression of *SLC2A3* (*GLUT3*), *SLC2A5* (*GLUT5*) and *SLC2A14* (*GLUT14*) compared to both bacteria (Supplementary Fig. 2). In contrast, most TCA cycle genes showed no change in expression or were downregulated following exposure to all pathogens at both timepoints (e.g., *IDH1*) (Fig. 1d).

Taken together, stimulation with both fungal and bacterial pathogens activated a number of immune response-relevant pathways that were distinguishable by the type of pathogen.

### reQTLs modulate the expression of 11% of regulated genes

To determine the influence of host genetic variability on pathogen-triggered transcriptional activation, we performed *cis* eQTL analysis for all stimulated and control conditions to identify SNPs that influence gene expression within a 1 Mb interval on either side of a transcript.

We identified 2718 to 3646 *cis* eQTLs in pathogen-stimulated monocytes (Supplementary Data 2). A comparison with recent data by Oelen et al. showed more than 80% overlap of their eQTL-regulated monocyte genes (504/617 *cis* eQTLs, 3 h stimulated monocytes) with our data (Supplementary Fig. 3), despite the use of different pathogens[7]. These correspond to only a fraction of all *cis* eQTLs detected in this study, which had 3929 additional eQTL-regulated genes after 3 h pathogen exposure.

We computed a beta-comparison and analyzed if the regression slopes of *cis* eQTLs under baseline and pathogen-stimulated conditions differed significantly[6]. The resulting reQTLs were defined as *cis*

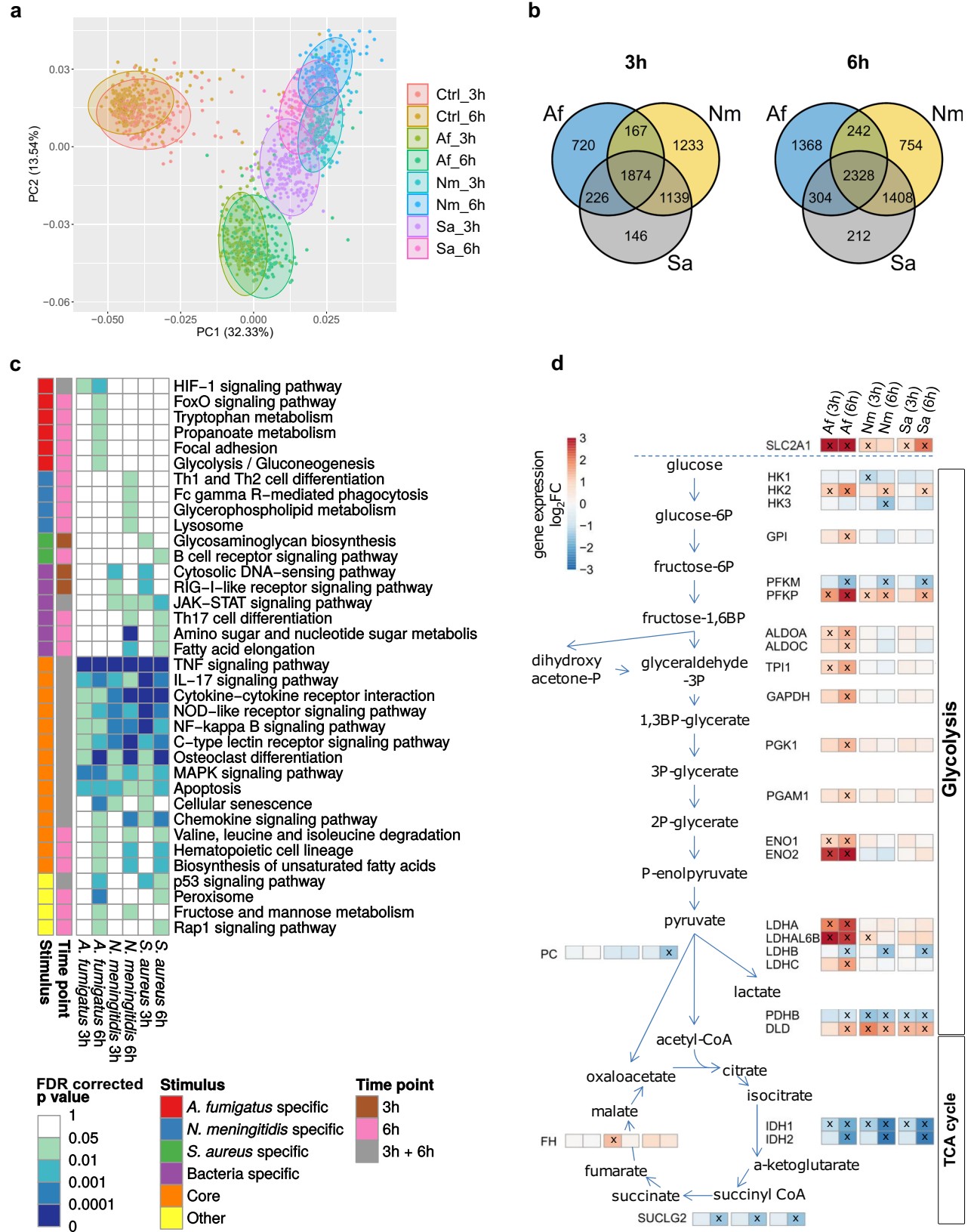

eQTLs with a significant difference in the regression slope of at least one stimulated monocyte condition compared to unstimulated monocytes. For each pathogen, we identified 6–12% of *cis* eQTLs as reQTLs (Fig. 2a). Of those, 10–18% were timepoint-specific with significantly different regression slopes for 3 h and 6 h of the same condition (Fig. 2b). Overall, we identified 745 reQTL-regulated genes,

which corresponds to 11% of all differentially expressed genes (Supplementary Data 3).

The fold-change (FC) distribution for all genes differed significantly from the FC distribution for reQTL-regulated genes (3 h: Fig. 2c, 6 h: Supplementary Fig. 4a). Specifically, comparing the FC distribution of all DEGs against reQTL-regulated genes with an

**Fig. 1 | Overview of the transcriptional immune response in human monocytes.**
**a** Principal component analysis of RNA-sequencing data of monocytes from 215 healthy volunteers. Each dot is one individual's transcriptional response in unstimulated monocytes (Ctrl) and after exposure to *A. fumigatus* (Af), *N. meningitidis* (Nm), or *S. aureus* (Sa) for 3 h and 6 h. Colors indicate stimulus and timepoint. **b** Venn diagrams of all differentially expressed genes (DEGs) at 3 h (left) and 6 h (right). Besides a pathogen-independent core immune response a substantial set of DEGs was specific for *A. fumigatus*. DEG sets induced by both bacteria were mostly overlapping. **c** Pathway enrichment analysis by overrepresentation analysis for DEGs of stimulated *vs.* unstimulated monocytes identified 36 relevant KEGG pathways (full list: Supplementary Data 1). False discovery rate (FDR)-corrected *p*-values are indicated. **d** Log$_2$ of fold-changes (log$_2$FC) of expression for genes encoding glycolysis and tricarboxylic acid (TCA) cycle enzymes. Significant differential gene expression was detected in at least one condition (marked with X). RNA-sequencing data are accessible through GEO accession number GSE177040. List of all DEGs and full list of enriched KEGG pathways are provided in Supplementary Data 1. Source data are provided as a Source Data file.

absolute log$_2$FC ≥ 1, reQTL-regulated genes were significantly more upregulated (3 h: Fig. 2c, 6 h: Supplementary Fig. 4a). The strongest effect was identified during stimulation with *N. meningitidis* (3 h, $p = 2.3 \times 10^{-14}$) with 45% upregulation for all DEGs compared to 79% for all reQTL-regulated genes. This result hinted to a primarily cell-activating genetic influence.

### reQTL-regulated expression differs for fungal and bacterial stimulation

Investigating the expression patterns of genes significantly regulated by reQTLs showed that both bacteria induced a similar reQTL-regulated expression pattern that differed from the reQTL-induced immune response after fungal exposure (Fig. 2d).

Both bacteria induced an overlap of 32% reQTL-regulated genes after 3 h and of 46% after 6 h (Fig. 2e). The overlap for *A. fumigatus*- and bacteria-induced reQTL-regulated genes was smaller (*A. fumigatus* and *N. meningitidis*: 18% after 3 h, 17% after 6 h; *A. fumigatus* and *S. aureus*: 26% after 3 h, 23% after 6 h) (Fig. 2e). However, when genes were significantly regulated both upon *A. fumigatus* and bacteria stimulation the same—and not independent—reQTLs seem to be active at the majority of loci (between 90.70% and 96.16%) (Supplementary Data 4). Although the pathogen-independent core response involved about one-third of all DEGs, only one-tenth of all reQTL-regulated genes were common among genes altered after stimulation with any of the three pathogens (12% after 3 h (57 core response reQTL-regulated genes out of a total of 464 reQTL-regulated genes); 13% after 6 h (70 core response reQTL-regulated genes out of a total of 547 reQTL-regulated genes)) (Fig. 2e). Following 3 h, approximately 20% of genes regulated by reQTLs emerged specifically after exposure to *A. fumigatus*, 36% after *N. meningitidis* and 10% after *S. aureus* (6 h: 24%, 26% and 8%, respectively) (Fig. 2e).

To analyze the impact of pathogen complexity on the number of reQTL-regulated genes, we compared our results for *N. meningitidis* (containing meningococcal LPS) with a reQTL study that used *Escherichia coli*-derived LPS as stimulus in a comparable setting[6]. *N. meningitidis* exposure induced more reQTL-regulated genes than LPS (after 6 h: *N. meningitidis*, 485 genes; LPS, 176 genes). reQTL-regulated genes for both conditions overlapped by only 6.6%, showing that *N. meningitidis* induced additional reQTL effects that were different from the exposure to LPS alone and purified LPS may trigger reQTL effects that were not observed for a complex Gram-negative bacterium (Supplementary Fig. 4b).

### reQTLs affect major PRR signaling cascades

Given the fundamental importance of PRRs for pathogen recognition, we screened KEGG pathways encoding PRR signaling pathways for reQTL-regulated genes (Supplementary Figs. 5–8).

The NOD-like receptor signaling pathway was significantly enriched for DEGs after all treatments (Fig. 1c). Within this pathway, we identified a reQTL for *NOD1* after 3 h exposure to *N. meningitidis* and 6 h exposure to both bacteria (rs55740347: $p_{Nm\_3h} = 3.88 \times 10^{-06}$, rs62447420: $p_{Nm\_6h} = 1.11 \times 10^{-10}$, rs4720003: $p_{Sa\_6h} = 8.44 \times 10^{-11}$, Supplementary Fig. 5). The strongest effect was identified after 6 h stimulation with *S. aureus*. Carriers of allele T showed increased *NOD1* expression over carriers of the C allele (Fig. 3a). NOD1 is an intracellular

sensor of bacterial peptidoglycan and induces recruitment of immune cells and complement factors early in the innate immune response[15]. We identified another reQTL for *CASP1* in the NOD-like receptor signaling pathway following 6h exposure to *A. fumigatus* (rs7934144: $p_{Af\_6h} = 2.07 \times 10^{-03}$, Fig. 3b, Supplementary Fig. 5). Carriers of allele G showed decreased *CASP1* expression compared to individuals with the opposite allele. Caspase-1, with inflammasome components, regulates processing of pro-interleukin (IL)−1β and pro-IL-18 to active IL-1β and IL-18, which are important in the host response to pathogenic microbes[16].

The C-type lectin receptor signaling pathway was also significantly enriched for DEGs after all treatments (Fig. 1c). *N. meningitidis* induced a reQTL for *PPP3CA* (or *CNA*) that encodes the catalytic subunit of calcineurin (rs17030831: $p_{Nm\_6h} = 3.58 \times 10^{-04}$, Fig. 3c, Supplementary Fig. 6). Calcineurin dephosphorylates the transcription factor NF-AT that regulates expression of immunomodulatory cytokines. Carriers of allele A showed decreased *PPP3CA* expression compared to carriers of another allele (Fig. 3c).

In the TLR signaling pathway, we identified a reQTL for *CD86* after exposure to *A. fumigatus* at both timepoints (rs3833583: $p_{Af\_3h} = 2.4 \times 10^{-02}$, $p_{Af\_6h} = 1.4 \times 10^{-02}$, Fig. 3d, Supplementary Fig. 7). Carriers of the allele A showed increased *CD86* expression compared to carriers of the opposite allele. This gene encodes a co-stimulatory molecule that is crucial in induction and regulation of adaptive immune responses.

In the complement cascade, the reQTL for *ITGAX* (or *CD11C*) showed reQTL effects in opposite directions after *N. meningitidis* exposure and stimulation with *A. fumigatus* and *S. aureus*. Following 6h exposure to *N. meningitidis*, SNP rs79258792 regulated expression of *ITGAX* ($p_{Nm\_6h} = 2.1 \times 10^{-10}$) and carriers with an ATAAA insertion showed elevated *ITGAX* expression levels compared to carriers of allele T (Fig. 3e, Supplementary Fig. 8). *A. fumigatus* and *S. aureus* had the opposite effect on *ITGAX* expression. *ITGAX* encodes for the integrin αX chain and forms together with integrin β2 (CD18) the complement receptor 4 (CR4, or CD11c/CD18), an integrin that is important for phagocytosis of complement-coated microbes[17].

In summary, our analyses identified a number of pathogen-specific reQTL-regulated DEGs important for immune responses and showed either stronger or even opposite reQTL effects for specific types of pathogens.

### reQTLs are functionally relevant in PRR signaling

Using ex vivo assays we examined the functional effect of two identified reQTLs on cytokine production using cells from nine female donors. For functional analysis of the *NOD1* rs62447420 SNP we isolated monocytes from freshly drawn blood and stimulated them with *S. aureus* and *A. fumigatus* to quantify activation-dependent IL-1β secretion. In accordance with results obtained in our reQTL study, monocytes from donors carrying the T variant of the *NOD1* rs62447420 secreted significantly more IL-1β compared to donors carrying the C variant of this SNP in response to pattern recognition of *S. aureus* (Fig. 4b), whereas only a slight effect could be observed after confrontation with *A. fumigatus* (Fig. 4c). Stimulation with LPS served as positive control for NOD1 receptor activation and showed a strong IL-1β release, especially in donors

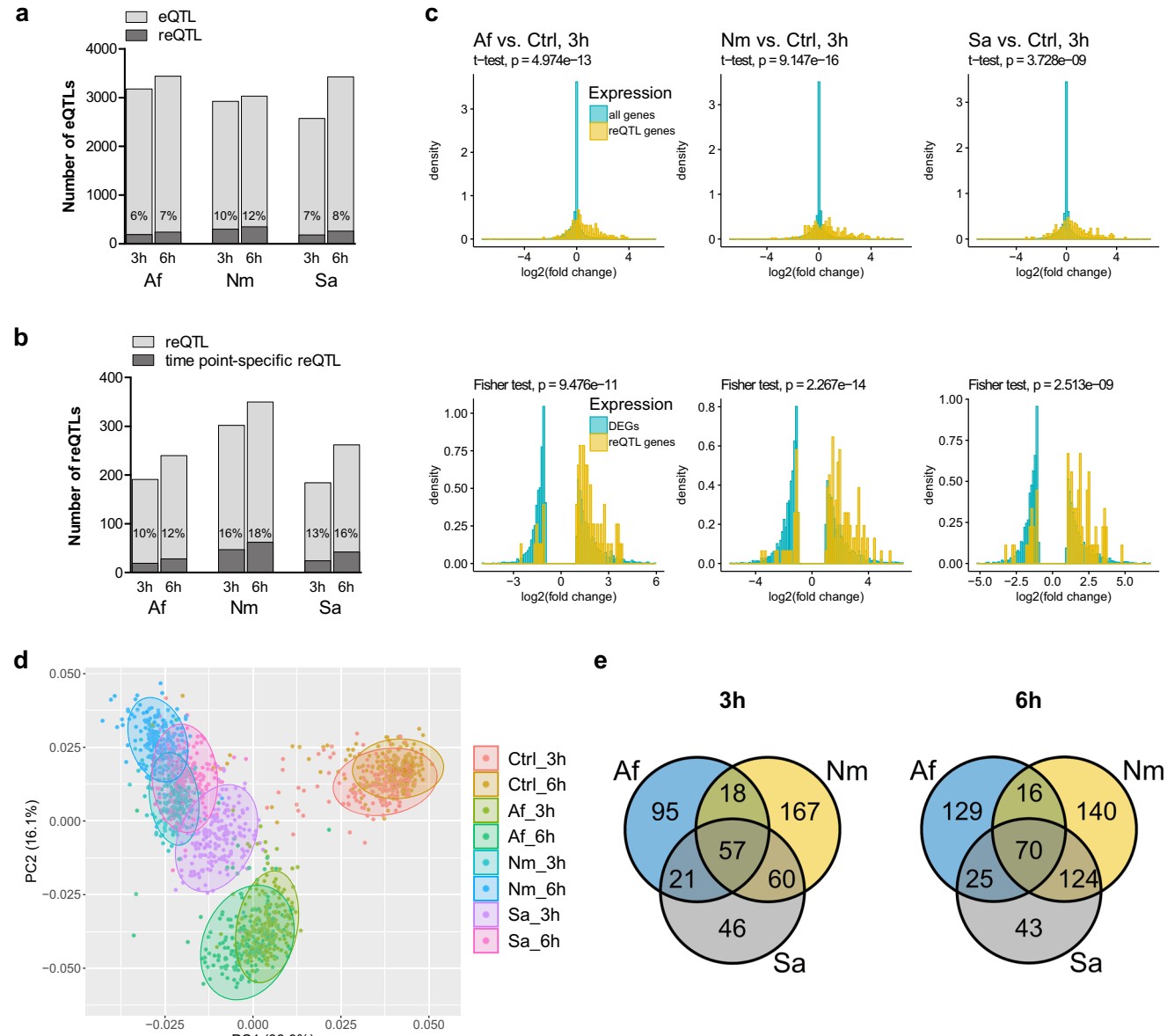

**Fig. 2 | Overview of reQTL effects in pathogen-exposed monocytes. a** Percentage of response expression quantitative trait loci (reQTLs) from all *cis* eQTLs in pathogen-exposed monocytes (*A. fumigatus* [Af]: 3,175 eQTLs and 191 reQTLs at 3 h, 3,439 eQTLs, and 240 reQTLs at 6 h; *N. meningitidis* [Nm]: 2,921 eQTLs and 302 reQTLs at 3 h, 3,029 eQTLs and 350 reQTLs at 6 h; *S. aureus* [Sa]: 2,574 eQTLs and 184 reQTLs at 3 h, 3,425 eQTLs, and 262 reQTLs at 6 h). **b** Percentage of timepoint-specific reQTLs in pathogen-exposed monocytes (Af: 19 at 3 h, 28 at 6 h; Nm: 47 at 3 h, 62 at 6 h; Sa: 24 at 3 h, 42 at 6 h). **c** Histograms comparing the gene expression log₂FC distribution of all genes (top row) or all DEGs (|log₂FC|≥ 1, bottom row) *versus* all reQTL-regulated genes after 3 h of stimulation (6 h stimulation: Supplementary Fig. 4a). Distribution differences were tested for statistical significance. Two-tailed *t*-test or Fisher exact test-based *p*-values are indicated. **d** Principal component analysis of 745 reQTL-regulated genes in monocytes of 215 healthy volunteers after exposure to Af, Nm, or Sa for 3 h and 6 h. Colors indicate stimulus and timepoint. **e** Venn diagrams of all reQTL-regulated genes after 3 h (left, total of 464 reQTL-regulated genes) and 6 h (right, total of 547 reQTL-regulated genes). 369 different genes are reQTL-regulated upon 3 h bacterial stimulation, 32% (117) of

them are reQTL-regulated following Nm and Sa stimulation (6 h: 46% (194 over-lapping reQTL-regulated genes out of 418)). 418 different genes are reQTL-regulated after 3 h stimulation with Af or Nm, 18% (75) of them are reQTL-regulated following Af and Nm stimulation (6 h: 17% (86 overlapping reQTL-regulated genes out of 504)). 297 different genes are reQTL-regulated after 3 h stimulation with Af or Sa, 26% (78) of them are reQTL-regulated following Af and Sa stimulation (6 h: 23% (95 overlapping reQTL-regulated genes out of 407)). 20% of genes are speci-fically reQTL-regulated after 3 h exposure to Af (95 out of 464 reQTL-regulated genes), 36% of genes are specifically reQTL-regulated after 3 h exposure to Nm (167 out of 464 reQTL-regulated genes) and 10% of genes are specifically reQTL-regulated after 3 h exposure to Sa (46 out of 464 reQTL-regulated genes) (6 h: 24% specifically reQTL-regulated genes after Af, 26% after Nm and 8% after Sa stimula-tion). *Ctrl* unstimulated, *DEGs* differentially expressed genes, *FC* fold-change. List of all DEGs is provided in Supplementary Data 1, while eQTL and reQTL data are provided in Supplementary Data 2 and 3, respectively. Source data are provided as a Source Data file.

carrying the T variant (Fig. 4a). We additionally assessed functional relevance of the reQTL effect on *CD86* gene expression that encodes a co-stimulatory molecule necessary for T cell activation. We isolated monocytes as well as T cells from freshly drawn blood and stimulated these with *S. aureus* and *A. fumigatus* to measure CD86-

driven T cell activation by the resulting IFNγ secretion. Carriers of the A allele showed a trend towards increased IFNγ levels in response to both pathogens, although results were not significant (Fig. 4e, f). T-cell activation was validated by stimulation with CD3 antibody (Fig. 4d).

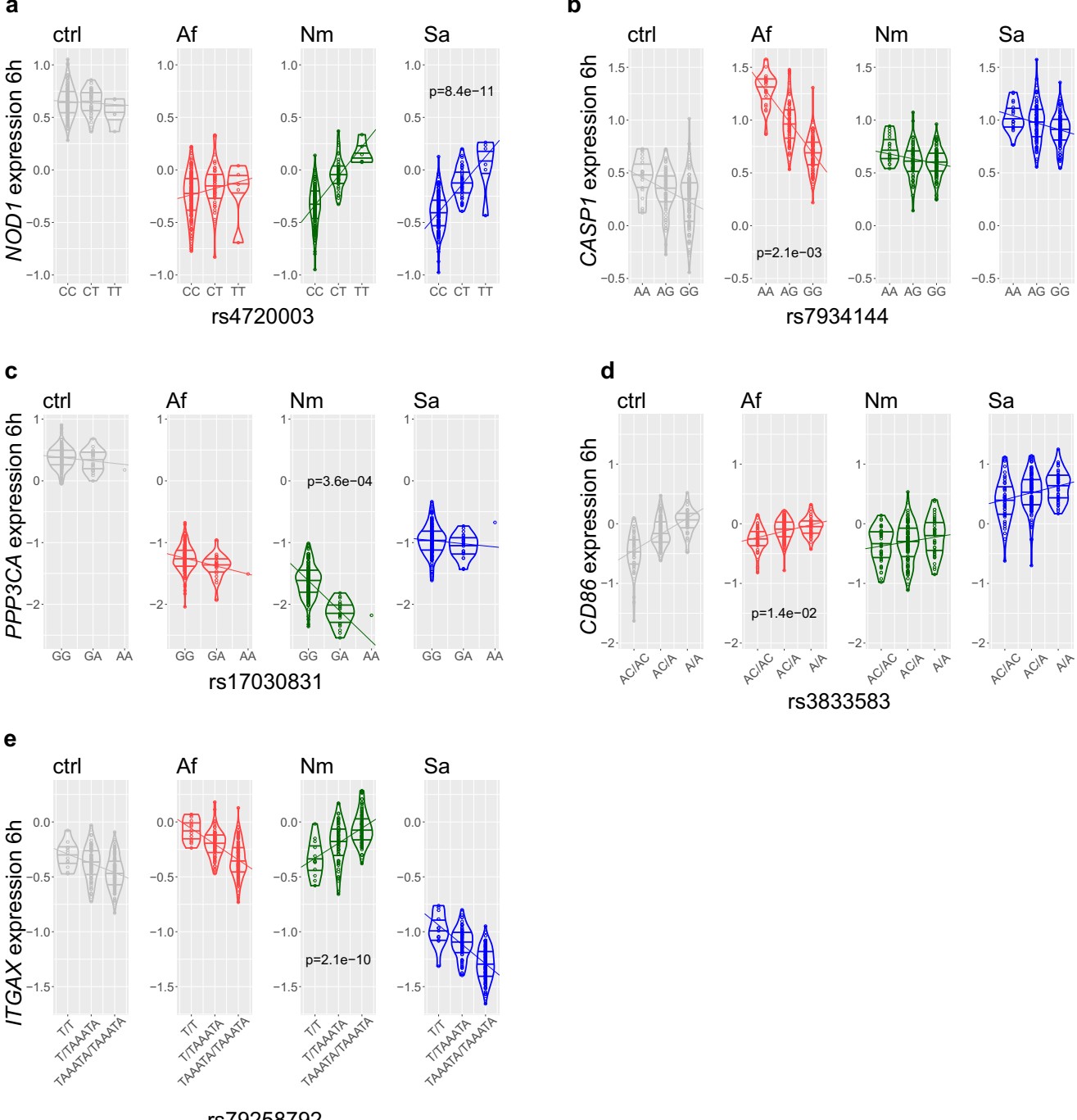

**Fig. 3 | reQTL-regulated genes that correspond to pattern recognition receptor signaling. a–d** Violin plots with allele-dependent differences in expression of *NOD1*, *CASP1*, *PPP3CA,* and *CD86* in *A. fumigatus* (Af), *N. meningitidis* (Nm) or *S. aureus* (Sa) stimulated monocytes following 6 h treatment. Distribution of centralized log$_2$ expression is shown for genotypes of indicated single-nucleotide polymorphisms (SNPs). Dashed lines: regression slope of gene expression values *vs.* genotype. Dots: individuals for the respective genotype. Horizontal lines in violins indicate 75th percentile, median and 25th percentile from top to bottom. For *NOD1* and *CD86*, only the top SNP for a condition with the strongest *p* value is shown. **e** Allele-dependent differences in expression of *ITGAX* with opposite effect directions after stimulation with *N. meningitidis* compared to *A. fumigatus* ($p_{Nm-Af} = 2.63 \times 10^{-17}$) and *S. aureus* ($p_{Nm-Sa} = 2.33 \times 10^{-22}$) for 6 h. Bonferroni corrected *p*-values from z test of differences in regression coefficients to unstimulated control (ctrl) are indicated. Ctrl: unstimulated. Source data are provided as a Source Data file.

Cells used for functional confirmation were isolated from female donors to additionally test the influence of the donor's sex. Although not tested for all initially identified reQTLs, our results indicate that their effects are likely not specific to one sex, but may underly a sex-independent immune response.

## Unique reQTLs pinpoint pathogen-specific immune function regulators

We screened for unique reQTLs to identify reQTL-regulated genes with inter-individual variability exclusively after exposure to a specific pathogen. reQTLs were defined by significant differences

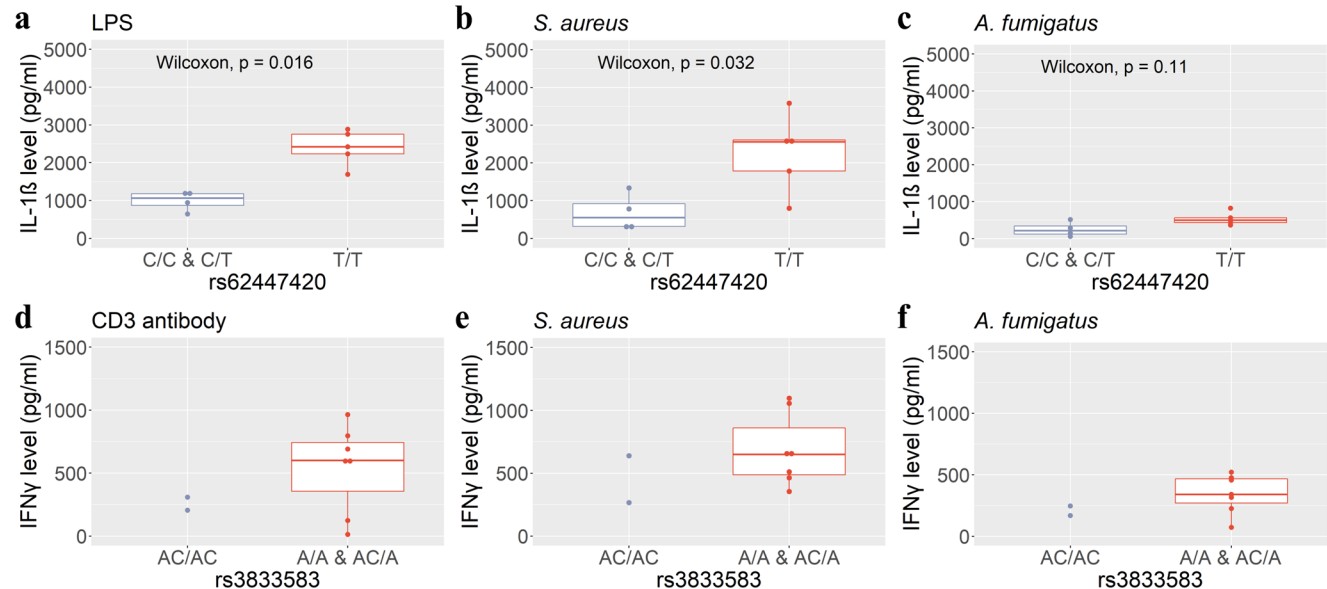

**Fig. 4 | Functional validation of reQTL effects on *NOD1* and *CD86* gene expression. a–c** IL-1β levels were analyzed after monocyte stimulation with **a** lipopolysaccharide (LPS), **b** *S. aureus* and **c** *A. fumigatus* in cells isolated from female donors with *NOD1* rs62447420 C/C & C/T ($n = 4$) and T/T ($n = 5$) genotypes. **d–f** T cell activation was measured by IFNγ release upon treatment with **d** CD3 antibody or during co-culture with monocytes isolated from female donors carrying *CD86* rs3833583 AC/AC ($n = 2$) or A/A & AC/A ($n = 7$) genotypes after

stimulation with **e** *S. aureus* and **f** *A. fumigatus*. Horizontal lines in box plots indicate 75th percentile, median and 25th percentile from top to bottom. Whiskers indicate the minimum and maximum values in 1.5 times the interquartile range. Dots represent individuals for the respective genotype. Two-tailed Wilcoxon rank sum test-based *p* values are indicated, except for **d–f**, where low sample size in AC/AC precluded statistical testing. Source data are provided as a Source Data file.

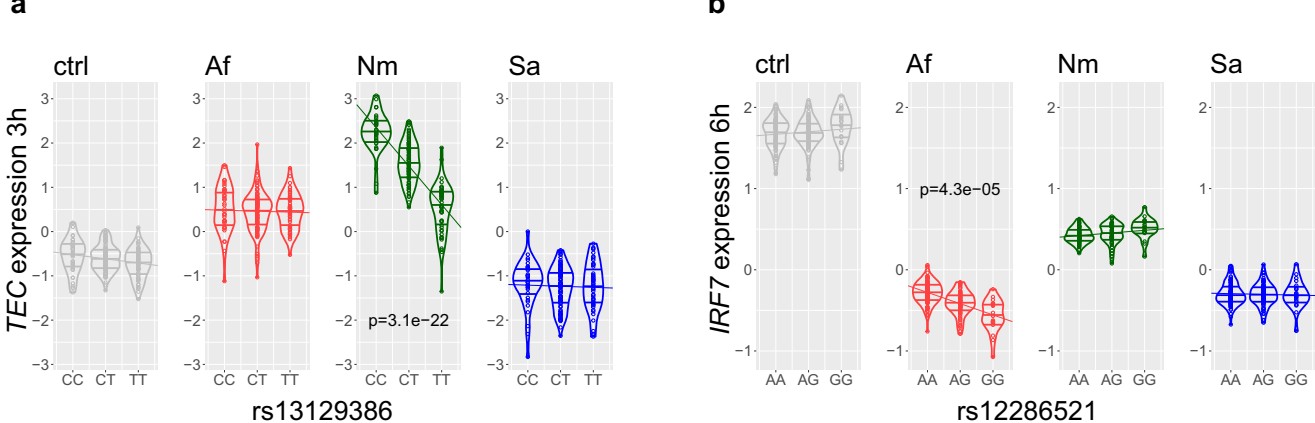

**Fig. 5 | Unique reQTLs for *N. meningitidis* and *A. fumigatus* stimulation. a** Allele-dependent differences in gene expression of *TEC*, the strongest unique response expression quantitative trait locus (reQTL) after 3 h *N. meningitidis* (Nm) stimulation. Differences in regression slope with *A. fumigatus* (Af): $p_{Nm\text{-}Af} = 4.41 \times 10^{-22}$ and *S. aureus* (Sa): $p_{Nm\text{-}Sa} = 1.98 \times 10^{-19}$. **b** Allele-dependent differences in expression of

*IRF7*, a unique reQTL-regulated gene for *A. fumigatus* exposure (6 h) with differences in regression slope for *N. meningitidis* ($p_{Af\text{-}Nm} = 6.5 \times 10^{-10}$) and *S. aureus* ($p_{Af\text{-}Sa} = 1.9 \times 10^{-04}$). Bonferroni corrected *p*-values from z test of differences in regression coefficients to unstimulated control (ctrl) are indicated. Ctrl: unstimulated. Source data are provided as a Source Data file.

in regression slopes under baseline and pathogen-exposed conditions. Unique reQTLs had the additional requirement of a significantly different regression slope for only one stimulus. This definition included no significant difference in regression slope in untreated monocytes or during stimulation with the other two pathogens. These criteria found 35 and 20 unique reQTLs, respectively, that became functionally active following *A. fumigatus* and *N. meningitidis* exposure. No unique reQTLs were found for stimulation with *S. aureus* (Supplementary Data 5).

The most significant unique reQTL was identified after 3 h exposure with *N. meningitidis* and influences the expression of *TEC* (rs13129386: $p_{Nm\_3h} = 3.08 \times 10^{-22}$, Fig. 5a). Carriers of the T

allele showed decreased *TEC* expression compared to carriers of the opposite allele. *TEC* encodes a nonreceptor tyrosine kinase that contributes to signaling from many receptors. Wang et al. showed that *TEC* is upregulated in macrophages upon LPS stimulation and blocking of Tec kinase led to decreased secretion of monocyte chemotactic protein (MCP)−1 and decreased expression of ICAM-1[18].

Exposure of monocytes to *A. fumigatus* for 6 h induced a unique reQTL regulating the *IRF7* gene (rs12286521: $p_{Af\_6h} = 4.27 \times 10^{-05}$, Fig. 5b). Carriers of the G allele showed decreased *IRF7* expression compared to carriers of the A allele. IRF7 is a transcription factor and regulates type-I interferon immune responses[19].

### reQTL analysis reveals potential mechanisms leading to non-communicable diseases

The majority of risk-SNPs for complex diseases identified via GWAS are in noncoding regions of the genome and have gene-regulatory functions[9,20]. However, identifying causal mechanisms and corresponding risk genes is nearly impossible by only relying on GWAS. eQTLs can be used to elucidate gene-regulatory mechanisms at GWAS-identified loci. reQTLs are particularly useful because they are from cells in a pathophysiologically relevant state. The reQTLs we identified may contribute to understanding the molecular mechanisms leading to diseases that are not caused by microbial infections.

To compare our reQTLs with GWAS findings, we identified all reQTLs with high LD ($r^2 > 0.8$) with a genome-wide significantly associated risk SNP for a disease in the GWAS catalog[21]. For consistency with our monocyte donors and to maximize power, only GWAS with European study cohorts and minimum 1000 cases per study were included. With these criteria, we found 60 distinct reQTL-regulated genes linked to disease-risk-SNPs (Supplementary Data 6). Consistent with the biological function of PRRs and DAMPs in sensing exogenous and endogenous risk patterns, the identified pairs of reQTLs and GWAS risk-SNPs were mainly linked to autoimmune and oncological diseases (Fig. 6a).

The most significant GWAS reQTL was rs10063083, which regulates the expression of the *NIPAL4* gene that encodes a plasma membrane Mg$^{2+}$ transporter. reQTL effects with significantly increased *NIPAL4* gene expression in carriers of allele C were induced by *A. fumigatus* at both timepoints and even more strongly by *N. meningitidis* following 6 h of stimulation (rs10063083: $p_{Af\_3h} = 2.7 \times 10^{-04}$, $p_{Af\_6h} = 8.3 \times 10^{-05}$, $p_{Nm\_6h} = 1.1 \times 10^{-23}$, Fig. 6b). The reQTL is in high LD ($r^2 = 0.90$) to rs6860540, which showed genome-wide significant association in a psoriasis GWAS[22]. In the GWAS, *ADAM19* was highlighted as plausible candidate gene at this locus because it is nearest to the GWAS risk SNP[22].

We identified a reQTL for *IRF8* in monocytes after 6 h stimulation with *N. meningitidis* and *S. aureus* (rs11117432: $p_{Nm\_6h} = 5.55 \times 10^{-14}$, $p_{Sa\_6h} = 2.67 \times 10^{-05}$, Fig. 6c). This reQTL was identical to the GWAS SNP associated with primary biliary cholangitis[23]. In the corresponding GWAS allele G was found to be disease-associated. Our data revealed that all carriers with the risk allele G showed increased *IRF8* expression compared to carriers with allele A. Furthermore, this reQTL is in LD to GWAS-SNPs associated with inflammatory bowel disease (rs11641016, $r^2 = 0.86$)[24], systemic lupus erythematosus (rs11117433, $r^2 = 1.0$)[25] and multiple sclerosis (rs17445836, $r^2 = 1.0$)[26]. *IRF8* encodes a transcription factor that regulates myeloid cell differentiation and expression of genes stimulated by IFN-α and IFN-β.

In the promotor region of *TNFRSF10A*, we identified a reQTL following exposure to all pathogens (rs13278062: 3 h: $p_{Af\_3h} = 9.52 \times 10^{-25}$, $p_{Nm\_3h} = 7.37 \times 10^{-28}$, $p_{Sa\_3h} = 8.81 \times 10^{-16}$, Fig. 6d; 6 h: $p_{Af\_6h} = 1.42 \times 10^{-07}$, $p_{Sa\_6h} = 3.71 \times 10^{-04}$). This reQTL was identical to rs13278062 that showed significant association in a GWAS for age-related macular degeneration with the disease-associated allele T[27]. In our study, the reQTL was linked to decreased *TNFRSF10A* expression in carriers of allele T compared to carriers of allele G. *TNFRSF10A* encodes a receptor that is activated by TRAIL1, which induces apoptosis[28].

Finally, we performed a colocalization analysis using coloc for all above-mentioned variants (Supplementary Data 7). We found evidence for colocalization for *IRF8* in monocytes after 6 h exposure to *N. meningitidis* and *S. aureus* with multiple sclerosis ($H_{4,Nm} = 0.997$, $H_{4,Sa} = 0.997$) and systemic lupus erythematosus ($H_{4,Nm} = 0.989$, $H_{4,Sa} = 0.990$), but only moderate evidence with inflammatory bowel disease ($H_{4,Nm} = 0.577$, $H_{4,Sa} = 0.616$). In contrast, under these conditions we found no evidence for colocalization with primary biliary cholangitis ($H_{4,Nm} = 0.140$, $H_{4,Sa} = 0.004$). However, this estimate is not based on the GWAS that reported this association[23], because corresponding data were not publicly available. For age-related macular

degeneration we found strong colocalization for *TNFRSF10A* in monocytes following 6 h exposure to *N. meningitidis* or *S. aureus* ($H_{4,Nm} > 0.999$, $H_{4,Sa} > 0.999$). Unfortunately, we could not test for colocalization for *NIPAL4* and psoriasis, because no GWAS summary statistic was publicly available for this trait, which showed association for the corresponding locus.

## Discussion

Genetic variation has a major impact on immune activation and thus influences susceptibility to infection, autoimmunity, and other diseases[29]. The identification of genetic polymorphisms linked to a gene expression phenotype (eQTLs) can provide explanations for individual differences in innate immunity. Importantly, while most studies so far focus on eQTLs in naïve cells, external activation can significantly alter expressional regulation, resulting in activity of reQTLs[4–6]. Given the central role of monocytes in many pathologies, the aim of this study was to investigate the impact of host genetic variation on their inter-individual immune response patterns to complex pathogens. Here we used a large sample cohort of 215 male individuals to assess the genetic regulation of immune activation in monocytes while minimizing background variability. Importantly, our data were generated with a highly purified monocyte preparation to exclude differential regulation in distinct cell types, which can preclude eQTL identification in mixed cell populations such as peripheral blood mononuclear cells (PBMCs)[30]. The use of purified immune cell populations is supported by the strong cell-type-specificity of the gene expression response and its genetic regulation in PBMCs upon pathogen stimulation. Among all peripheral blood immune cells, monocytes are the most promising target due to a higher number of DEGs and reQTLs compared to other cell types[7].

Analysis of pathogen-induced gene expression in monocytes identified a core expressional pattern encompassing approximately one-third of DEGs. Beyond this core response, expression patterns enabled a clear discrimination between fungal and bacterial stimulation. In contrast, transcriptional adaptation of the monocytes to both bacterial stimuli showed high similarities, overlapping up to 71%. These ex vivo data mirror in vivo findings like discovery of a specific transcriptional monocyte state as a signature in bacterial sepsis, enabling discrimination of bacterial sepsis from sterile inflammation[31]. In addition, our data also confirm and extend prior studies, showing that transcriptional marker sets enable discrimination of bacterial and fungal infection in an ex vivo human whole-blood infection model[32]. Interestingly, specificity was even higher with regard to genetic regulation: Only about one-tenth of all reQTL-regulated genes were found for all three pathogens. Pathogen-induced reQTLs regulated immune cell-activating gene expression and affected predominantly upregulated genes. reQTLs influence important PRR signaling pathways, such as NOD-like receptor signaling (*NOD1*, *CASP1*), C-type lectin receptor signaling (*PPP3CA*), TLR signaling (*CD86*), and complement and coagulation cascades (*ITGAX*)[33,34]. Thus, these data provide a functional explanation for individual differences in innate response patterns and demonstrate that genetic polymorphisms are important in determining the transcriptional response to infection in human monocytes.

In order to enhance the probability for detection of eQTL- and further reQTL-affected gene expression we initially screened exclusively male donors. Although this allowed us to control background variation it precluded generalization of our findings at this point. Indeed, biological sex is associated with major differences in immune cell gene expression[35]. Furthermore, it is uncertain how much additional variance of the immune response is introduced by cyclical changes of sex hormones in women[36]. Despite these reported sex-associated differences, we were able to confirm selected reQTLs for female donors, showing that at least some important reQTLs identified in this study are also active in a female background. The reQTL for *NOD1* in monocytes was identified after stimulation with *N.*

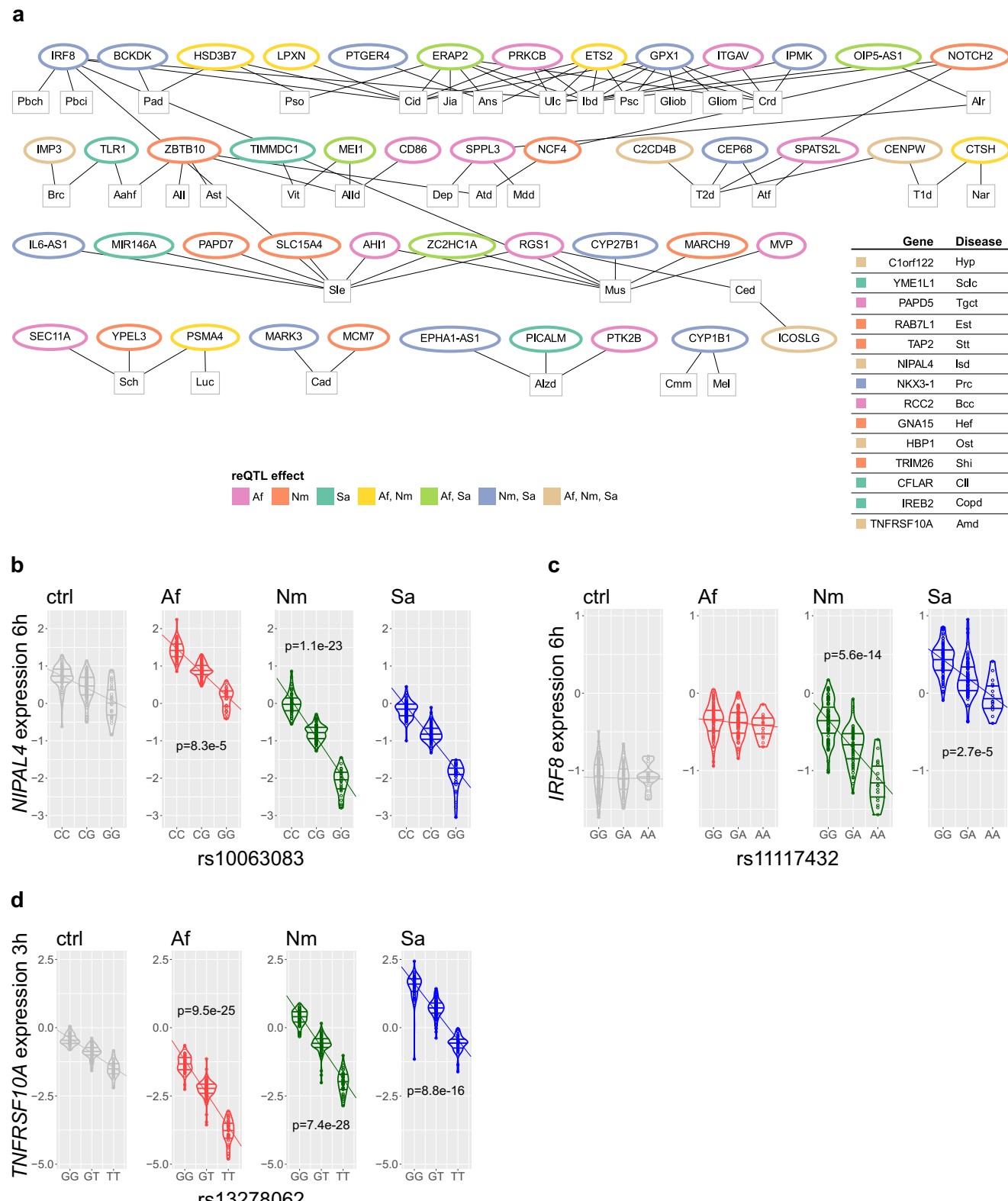

**a**

reQTL effect

Af | Nm | Sa | Af, Nm | Af, Sa | Nm, Sa | Af, Nm, Sa

| Gene | Disease |
|---|---|
| C1orf122 | Hyp |
| YME1L1 | Sclc |
| PAPD5 | Tgct |
| RAB7L1 | Est |
| TAP2 | Stt |
| NIPAL4 | Isd |
| NKX3-1 | Prc |
| RCC2 | Bcc |
| GNA15 | Hef |
| HBP1 | Ost |
| TRIM26 | Shi |
| CFLAR | Cll |
| IREB2 | Copd |
| TNFRSF10A | Amd |

**b** *NIPAL4* expression 6h — rs10063083 (ctrl, Af, Nm, Sa); p=8.3e-5 (Af), p=1.1e-23 (Nm)

**c** *IRF8* expression 6h — rs11117432 (ctrl, Af, Nm, Sa); p=5.6e-14 (Nm), p=2.7e-5 (Sa)

**d** *TNFRSF10A* expression 3h — rs13278062 (ctrl, Af, Nm, Sa); p=9.5e-25 (Af), p=7.4e-28 (Nm), p=8.8e-16 (Sa)

*meningitidis* and *S. aureus*. *NOD1* encodes an intracellular PRR that drives proinflammatory and antimicrobial responses[15]. Functional relevance of this reQTL in PRR signaling was validated by stimulation of NOD1 signaling in primary monocytes with *S. aureus*, showing an increased cytokine release by donors carrying the allele associated with higher *NOD1* expression. In addition, functional reQTL effect was shown for *CD86* upon *A. fumigatus* and *S. aureus* exposure. *CD86* encodes a co-signaling molecule on the surface of antigen-presenting cells that controls the T cell response to antigens in conjunction with T cell receptor signals, inducing adaptive immune responses[37]. Both CD86 and its paralog CD80 were upregulated in a model of experimental allergic aspergillosis, demonstrating a significant role for these co-stimulatory molecules in onset, persistence, and progression of immune responses[38]. Interestingly, this reQTL is in linkage disequilibrium (LD, $r^2 = 0.81$) with rs75557865, a risk SNP in GWAS for allergic disease (asthma, hay fever or eczema)[39]. The reQTL effect was

**Fig. 6 | reQTLs and regulated genes that correspond to disease-associated GWAS-SNPs. a** Disease-associated genome-wide association study single-nucleotide polymorphisms (GWAS-SNPs) in high linkage disequilibrium (LD) ($r^2 > 0.8$) with response expression quantitative trait loci (reQTL) in pathogen-exposed monocytes. reQTL-regulated genes are connected to diseases by lines. Inserted table lists reQTL-regulated genes connected to only one disease. Colors: exposure to different pathogens. **b**–**d** Allele-dependent differences in expression of *NIPAL4*, *IRF8* and *TNFRSF10A* following pathogen exposure. Corresponding reQTLs are in LD with GWAS-SNPs for **b** psoriasis, **c** primary biliary cholangitis, inflammatory bowel disease, systemic lupus erythematosus, multiple sclerosis or **d** age-related macular degeneration. For *IRF8* and *TNFRSF10A* only the top SNP for one condition with strongest *p*-value is shown. Ctrl: unstimulated, Af: *A. fumigatus*, Nm: *N. meningitidis*, Sa: *S. aureus*. Bonferroni corrected *p* values from z test of differences in regression coefficients to unstimulated control (ctrl) are indicated. List of disease-associated GWAS-SNPs with reQTL effects in stimulated monocytes with all associations of reQTLs and GWAS diseases and disease abbreviations is provided in Supplementary Data 7. Source data are provided as a Source Data file.

also reported in a study on IFNγ-stimulated monocytes[5], showing the robustness of the reQTL effect on *CD86*. Thus, this reQTL is a possible candidate for functional consequences for allergic diseases. We showed that part of our findings can be confirmed with female-derived samples, which allowed us to conclude that sex-independent reQTL effects exist. Generalizing on our findings, however, warrants further equally sized study cohorts including both sexes.

Another potential limitation of our study is the use of inactivated pathogens. We decided for this experimental approach to enhance comparability between the stimulations and avoid cell death during the co-incubation. Importantly, inactive pathogens will occur during natural infection and contribute to immune activation. Of note, using inactive pathogens provided a major increase in complexity compared to previous studies that assessed the effect of defined receptor-specific stimuli[5,6]. A comparison of the genetic regulation of monocyte genes in response to *N. meningitidis* (this study) *versus* a receptor-specific stimulus (*E. coli* LPS, Kim-Hellmuth et al.)[6] showed an overlap of less than 10%, although both stimuli induced a similar proportion of reQTLs (12% after *N. meningitidis*, 17% after LPS) for all eQTLs.

The potential impact of functional links between host genetic polymorphisms and gene expression identified in our study is potentially relevant for a range of clinical conditions associated with inflammation and immune activation. We combined our data with genetic variants in the GWAS catalog to link risk variants for autoimmune, inflammatory, and infectious diseases and cancer reQTL-regulated genes to find potential pathophysiological mechanisms[21]. The strongest reQTL that was linked to an association study was in LD with a risk variant for psoriasis and influenced *NIPAL4* expression. Mutations in *NIPAL4* genes are described in patients with congenital ichthyosis, a monogenic disease with cutaneous manifestation but unknown underlying pathogenetic mechanisms[40]. Gene expression profiling of skin from patients with congenital ichthyosis identified significant upregulation of psoriasis hallmark genes, while Nipal4-knockout mice exhibited neonatal lethality due to skin barrier defects[41,42]. A disease-associated SNP from GWAS for primary biliary cholangitis, inflammatory bowel disease, systemic lupus erythematosus, and multiple sclerosis was in strong LD to a reQTL that regulated expression of *IRF8*. Irf8, a key factor for dendritic cell, monocyte, and macrophage maturation, was described to influence the genetic risk for susceptibility to several chronic inflammatory diseases[43–45]. siRNA-mediated inhibition of Irf8 in vitro and in vivo reduces the intensity of the inflammatory response (mediated by TNF, IL-6, IL-12/IL-23, and IL-1β) and the associated pathology in a mouse model of intestinal colitis. Finally, a reQTL that was linked to a risk SNP for age-related macular degeneration was found to regulate *TNFRSF10A* gene expression. The binding of TRAIL to the encoded death receptor TRAILR1 is known to induce apoptosis[28]. TRAILR1 is present in cultured retinal pigment epithelium cells and late manifestation of age-related macular degeneration is characterized by atrophy of the retinal pigment epithelium, followed by degeneration of the choriocapillaris[46]. Our data showed that the risk allele for age-related macular degeneration is linked to increased *TNFRSF10A* expression compared to the opposite allele. Hence, our data indicate known (*IRF8*) as well as potentially new plausible explanations for diseases exemplified by *NIPAL4* association to skin disease or *TNFRSF10A* expression as possible pathophysiological mechanism inducing

increased apoptosis rate and thus influencing the course of age-related macular degeneration.

In summary, we provide a systematic characterization of genetic polymorphisms that directly impact gene expression in monocytes after activation by pathogens. Our data highlight a high degree of pathogen-specificity for genetic regulation when compared to the overall transcriptomic response. reQTLs mainly affected immune cell activation and were predominantly associated with upregulated genes. reQTL-regulated genes have central roles in PRR signaling pathways and provide plausible candidate genes for disease-associated variants. Our joint analysis of the pathogen-driven transcriptional immune response and its dependence on genetic variation provides a valuable dataset to dissect inter-individual differences in the response towards infection. At the same time, our data provides functional hypotheses for a spectrum of non-communicable diseases that have in the past been linked to defined genetic polymorphisms.

## Methods

### Study design

We performed transcriptomic analysis of monocytes from 215 individuals that were stimulated with *A. fumigatus*, *N. meningitidis*, and *S. aureus* or left unstimulated for 3h and 6h. Each participant was genotyped. Linear regression analysis was used to identify eQTLs and stimulus-induced reQTLs (Supplementary Fig. 1).

### Donor collection

This study was approved by the Ethics Committees of the Friedrich-Schiller-University Jena (permit number: 3811-07/13) and the Julius-Maximilians-University of Würzburg (permit number: EK 191/21).

Venous blood from 215 healthy male volunteers of European ancestry was taken after written informed consent. Characteristics of included participants were age between 18 and 40 years (mean 28), non-smokers, C-reactive-protein ≤10 mg/l, without infection or vaccination for at least 4 weeks before blood withdrawal, and blood count within normal range. All inclusion criteria, including the restriction to male participants, were introduced to reduce confounding effects e.g., of sex- and age-specific differences. Supplementary Fig. 9 depicts the power analysis of our samples for the identification of eQTL effects under the assumption of different minor allele frequency (MAF) and effect sizes. The power analysis was performed with powerEQTL v0.3.4[47]. Fixed parameters were set as follows: sigma.$y$ = 0.13, nTests = 30,000,000.

For functional NOD1 and CD86 ex vivo assays, venous blood was collected from 9 healthy female volunteers after written informed consent. All donors were aged between 20 to 40 years (mean 28) without infection or vaccination for at least 4 weeks before blood withdrawal. These inclusion criteria were used to confirm the effect of selected reQTLs identified in male participants in a female background.

### CD14+ monocyte purification

CD14+ monocytes were isolated from EDTA-anticoagulated blood after density gradient centrifugation using Biocoll separating solution (Bio&Sell Cat# BS. L 6115) by magnetic-activated cell sorting using CD14 monoclonal anti-human antibody conjugated to Microbeads

(Miltenyi Biotec Cat# 130-050-201, RRID:AB_2665482) and the auto-MACS Pro Separator system according to the manufacturer's instructions (Miltenyi Biotec). A purity of >95% was determined by flow cytometry using a PerCP-labeled monoclonal anti-CD14 antibody (mouse IgG1, clone 47-3D6, Abcam Cat# ab91146, RRID:AB_10675401). CD14$^+$ monocytes were resuspended in RPMI1640 with L-glutamine (Bio&Sell Cat# BS. FG 1215) supplemented with 20% heat-inactivated standardized fetal bovine serum (FBS Superior (FBS), Bio&Sell Cat# BS. S 0615) and cultured at a density of $2.5 \times 10^6$ cells/ml. After overnight resting, monocyte viability was determined by trypan blue staining (≥85%) prior to stimulation. Heat inactivation of FBS was performed at 56 °C for 30 min.

## Stimulation of CD14$^+$ monocytes
Monocytes were stimulated with inactivated *A. fumigatus* germ tubes (multiplicity of infection [MOI] = 0.5), *S. aureus* (MOI = 5), *N. meningitidis* (MOI = 1) or media as unstimulated control for 3 h and 6 h. We used inactivated pathogens to enhance comparability between the stimulations and avoid cell death during the co-incubation. Different germ tube lengths of *A. fumigatus* or differences in density in pre-cultures may influence recognition by immune cells. To ensure a comparable immune response towards the pathogens, we used pathogen-specific MOIs. Expression of cell-surface activation markers and monocyte death rate upon pathogen exposure were used to adjust the MOI for each pathogen with the aim to induce a strong immune response with low cell death rate.

After stimulation, the cells were separated from the supernatant by centrifugation ($300 \times g$ for 10 min, 4 °C). Monocytes were lysed in Buffer RLT (Qiagen Cat# 79216). Lysates and supernatants were stored at −80 °C.

## Strains and culture
*Aspergillus fumigatus* ATCC 46645 was cultivated on Aspergillus-Minimal-Media-Agar for 5d at 37 °C and 5% CO$_2$[48]. Conidia were harvested in double-distilled H$_2$O and separated from mycelia by using MACS SmartStrainers (30 μm, Miltenyi Biotec Cat# 130-098-458). For induction of germ tubes, $4 \times 10^6$ conidia/ml were incubated for 5 h to 6 h in RPMI1640 with L-glutamine until at least 80% of the conidia showed germ tube formation.

*Neisseria meningitidis* serogroup C (WUE2120) was cultivated on blood agar plates at 37 °C and 5% CO$_2$ for 2d followed by transfer of the cells into proteose-peptone-media, supplemented with MgCl$_2$ (0.952 g/ml), NaHCO$_3$ (0.425 g/ml) and PolyVitex (10 ml/l, bioMérieux Cat# 55651) at 37 °C and 5% CO$_2$[49]. Cultures were harvested when exponential growth phase (OD$_{600}$ 0.6–0.7) was reached.

*Staphylococcus aureus* ATCC25923 was cultivated overnight in lysogeny broth (LB) medium (10 g/l tryptone, 5 g/l yeast extract, 10 g/l sodium chloride, pH7) at 37 °C to stationary phase, reseeded in LB medium and grown at 37 °C until the bacterial cells reached the exponential growth phase (OD$_{600}$ 0.6–0.7).

## Inactivation of strains
*A. fumigatus* and *N. meningitidis* were incubated for 1.5 h at 37 °C in PBS (w/o Ca$^{2+}$, Mg$^{2+}$) containing 0.1% thimerosal (Sigma-Aldrich Cat# T8784) for *A. fumigatus* and 0.05% for *N. meningitidis*. Cells were washed five times with PBS (w/o Ca$^{2+}$, Mg$^{2+}$, Sigma-Aldrich Cat# D8537) and resuspended in RPMI1640 with L-glutamine and 20% heat-inactivated FBS. Aliquots of $1.25 \times 10^7$ germ tubes/ml for *A. fumigatus* and $2.5 \times 10^7$ cells/ml for *N. meningitidis* were stored at −20 °C.

*S. aureus* was inactivated in 50% ethanol for 4 h at 37 °C and washed five times with PBS (w/o Ca$^{2+}$, Mg$^{2+}$) supplemented with 20% heat-inactivated FBS. The final concentration of $1.25 \times 10^8$ *S. aureus* cells/ml was adjusted in RPMI1640 with L-glutamine and 20% heat-inactivated FBS, aliquoted and stored at −20 °C.

Success of inactivation was confirmed for all three pathogens by plating.

## DNA and RNA isolation
DNA and RNA were purified using the AllPrep96 DNA/RNA Kit from Qiagen (Cat# 80311) according to the manufacturer's instructions. RNA and DNA samples were checked for quantity and quality by a NanoDrop 8000 spectrophotometer (Thermo Fisher Scientific). Subsequently, the RNA integrity (RIN > 8) was analyzed using the Agilent 2100 Bioanalyzer (Agilent). For RNA library preparation the RNA concentrations were fluorometric determined using the Invitrogen Quant-iT RNA Assay Kit (Thermo Fisher Scientific Cat# Q10213) and a TECAN Infinite microplate-reader (Tecan).

## Genotyping and imputation
Genotyping was conducted with the Illumina GSA1.0 Chip (Illumina Cat# 20030770) comprising 700,078 SNPs for 215 individuals. Initial quality control was performed with PLINK 1.9[50]. Samples with genotype call rate >97%, discrepancies in sex, divergent ancestry from the CEU HapMap[51] population and related samples were excluded from further analysis. All unambiguous SNPs with SNP call rate >95%, minor allele frequency (MAF) > 1%, and Hardy–Weinberg equilibrium (HWE) > 0.001 were used for imputation. Imputation was performed with IMPUTE2 (v2.3.2)[52] based on 1000 Genomes Phase 3 as reference panel. As post-imputation quality control, we excluded all variants with an information score <0.8, HWE < 0.001, MAF < 1%, and SNP missing rate >5% for best-guessed genotypes at posterior probability >0.9 from further analysis.

## RNAseq and gene expression analysis
RNA-sequencing used the QuantSeq-3′-mRNA library preparation kit (Lexogen Cat# 015.96). Next-generation sequencing (NGS) libraries were sequenced at the end of the 3′ poly-A tail using a HiSeq 2500 platform (Illumina, San Diego, USA). Quality control of the generated fastq files was performed with FastQC (v0.11.7). For adapter trimming bbduk from the BBMap (v37.44 https://sourceforge.net/projects/bbmap/) was used. The read alignment against GRCh38 primary alignment was performed with STAR Aligner (v2.5.2b). FeatureCounts (v1.20.6) was used for quantification of gene expression using the Ensembl annotation GRCh38.89 as reference[53]. All parameters were as recommended by the Lexogen integrated Data Analysis Pipeline guide. Summaries of all steps described were aggregated and reviewed with multiQC (v1.6)[54].

Pairwise tests were performed between control and stimulation for each timepoint or between stimulations per timepoint. Count matrices were normalized using median-by-ratio normalization (MRN). Four statistical tools (DESeq 1.30.0, DESeq2 1.18.1, limma voom 3.34.6, and edgeR 3.20.7) were used to report multiple test-corrected *p* values (Supplementary Fig. 10, Supplementary Data 8). In addition, mean MRN, transcripts per kilobase million, and reads per kilobase million values were computed per pairwise test including the corresponding log$_2$ of fold-changes. Gene expression differences were considered significant if they were reported significant by all four tools (*q* < 0.01 and |log$_2$(fold-change)| ≥1).

Pathway activity was analyzed by conducting KEGG pathway enrichment analysis with enrichR (v2.1.0)[55]. Gene expression data were further overlayed on KEGG pathway maps by using pathview (v1.26.0, R 3.6.3). R (v3.6.3) was used when R environment was required.

RNA-sequencing data are accessible through GEO Series accession number GSE177040.

## eQTL analysis
Quality control was performed for all samples and all conditions. Gene expression data were normalized using the R package DESeq2 (v1.18.1)

using size factors for all samples and all conditions. The read counts were log-transformed with an offset of 1. Only genes with log-transformed value greater 1 in at least 10% of samples were selected. Subsequently, all read counts were centered and distributed equally. All samples with a D statistic (D statistic described in: Wright et al.[56]) outside of 1.5*interquartile range (IQR) and less than three million mapped reads were excluded. Genes with greater or equal than 6 reads in at least 20% of samples were used for further analysis. All expression values between samples were trimmed mean of values normalized using R package edgeR (v3.20.7). For each gene, the gene expression was inverse-normal transformed over all samples to increase power. Annotation was performed using R package biomaRt (v2.36.1)[57]. Only genes on autosomes were included.

Expression quantitative trait loci (eQTL) analysis used QTLTools (v1.3.1)[58] for all four conditions (unstimulated, stimulated with *A. fumigatus*, *N. meningitidis* or *S. aureus*) and two timepoints (3 h, 6 h). *Cis* eQTLs (window size 1 MB) were calculated for 12,575 genes and 3,584,731 single-nucleotide polymorphisms (SNPs) using 30 PEER factors[59] and three genotyping principal components. Identification of eGenes (one eQTL per gene within each condition) was conducted according to Kim–Hellmuth et al.[6] using the permutation pass of QTLTools with the setting "−permute 1000." Permutation *p* values were gained by a beta approximation to correct for testing multiple SNPs as *cis* eQTL at each locus (local correction). Again within each condition, we then applied the false discovery rate (FDR) procedure by Storey and Tibshirani[60] using the R-package *q* value (v2.12.0) to correct genome-wide for testing multiple genes (global correction). Over all eight conditions we found 25,310 *cis* eQTLs where 17,851 *cis* eQTLs were distinct SNP-gene pairs. This refers to 6865 different eGenes and shows that the majority of *cis* eQTLs are present across different conditions.

## reQTL analysis

To detect response eQTLs (reQTLs), we used the most significant *cis* eQTLs under stimulation and compared the regression coefficients between baseline and stimulus to test significant differences in effect size of *cis* eQTLs in a z test as described in Kim-Hellmuth et al.[6]. Significant reQTLs were determined by correcting the *p* values from the z test for multiple testing using Bonferroni correction resulting in 1529 reQTLs where 1179 were distinct reQTLs from all six conditions (*A. fumigatus*, *N. meningitidis* and *S. aureus*, each 3 h and 6 h stimulation). Of note, previous studies (e.g., Kim et al.)[8] have used differential expression upon stimulation as a quantitative trait for the detection of reQTLs (called diffQTL). However, in our study by Kim-Hellmuth et al.[6] we have also calculated diffQTLs for all reQTLs identified by β-comparison and used Spearman correlation as a measure of similarity. This showed that both methods lead to nearly identical results.

For the identification of timepoint-specific reQTLs, we used all significant reQTLs from one timepoint and tested these against the other timepoint in a z test and vice versa. All reQTLs with a Bonferroni corrected *p* value < 0.05 were timepoint-specific.

Differences in the regression coefficients between the treatments were analyzed by testing the significant reQTLs of one treatment against the other two treatments of the same timepoint and vice versa. The resulting *p* values were Bonferroni corrected. Significant treatment-specific reQTLs had a corrected *p* value < 0.05.

Unique reQTLs for one stimulus were identified if only one condition showed a significant *cis* eQTL and no significance in the other conditions.

Finally, we tested the robustness of the stimulus-dependent reQTLs. For this, we used all 3279 transcripts that are present in 3 h and 6 h unstimulated monocytes. Of them, only 6 transcripts (0.2%) showed significant results when we compared the regression coefficients at both timepoints with the z test. The data showed that the

reQTLs in our study were almost entirely due to the stimulations and were barely artificially induced.

## Functional NOD1 and CD86 ex vivo assays

PBMCs were isolated from freshly drawn blood from a genetically pre-screened female donor cohort by Biocoll density gradient centrifugation. DNA from PBMCs was purified using the Roche High Pure PCR Template Preparation Kit (Sigma-Aldrich Cat# 11796828001) according to the manufacturer's instructions. Isolated DNA was then amplified (5´: TCACAGAAACCAGTGCCCAAGT, 3´: ATCATTGAAATTAATTT CACCTGTCCA) with 2x Q5 polymerase master mix according to the manufacturer's instructions (NEB Cat# M0492L). PCR samples were purified with the PCR & Gel Clean-Up Kit (Macherey-Nagel Cat# 740609.50) according to the manufacturer's instructions and sent to LGC genomics for sequencing with the 5´ primer. SNP genotypes were evaluated with ApE (v.3.0.8).

CD14+ monocytes were isolated from freshly isolated PBMCs by magnetic-activated cell sorting using human CD14 MicroBeads and the MidiMACS Separator system (Miltenyi Biotec). CD14+ monocyte viability of >99% was determined after isolation with the LUNA automated cell counter (Logos Biosystems). CD14+ monocytes were cultured and stimulated with *A. fumigatus* and *S. aureus* for 12 h under same conditions as mentioned above. Afterwards, IL-1β levels in supernatants were measured by ELISA according to manufacturer's instructions (Invitrogen Cat# BMS224-2). The standard curve was calculated from blank-curated mean standard values with a 4-parameter curve fit (R package dr4pl, v2.0.0) according to ELISA Kit manufacturer's instructions. T cells were isolated from CD14+ monocyte-cleared PBMC suspension by negative selection using the human Pan T Cell Isolation kit (Miltenyi Biotec Cat# 130-096-535). T cell viability of >99% was determined after isolation with the LUNA automated cell counter. T cells were cultured together with monocytes in a 9:1 ratio and stimulated with *A. fumigatus* and *S. aureus* for 12 h under same conditions as mentioned above. IFNγ levels in supernatants were measured by ELISA according to manufacturer's instructions (Invitrogen Cat# EHIFNG2). The standard curve was calculated from blank-curated mean standard values with standard curve fitting according to ELISA Kit manufacturer's instructions.

Monocyte and T cell stimulations as well as ELISA measurements were performed in duplicates for all samples. All sample values were blank-curated and medium control-curated before concentration calculation via the corresponding standard curve formula. Statistical analysis of IL-1β and IFNγ levels was performed with two-tailed Wilcoxon rank sum test.

## Overlap between reQTLs and GWAS catalog

The overlap between reQTLs and trait-associated variants was determined using the results of the NHGRI-EBI GWAS Catalog[21] (downloaded 01/30/2019 version v1.0). We selected for GWAS that analyze disease associations in European study cohort of >1000 people. A listed GWAS SNP was assumed to coincide with a reQTL if the GWAS SNP was in high linkage disequilibrium ($r^2 > 0.8$, human assembly GRCh37) with the top SNP per gene. A full list of the identified GWAS reQTLs is in Supplementary Data 6. Finally, we performed a colocalization analysis using coloc v5.1.1[61] for the GWAS reQTL pairs that are presented in the result section. For this, we used the summary statistics for the corresponding diseases−with the exception of psoriasis and age-related macular degeneration−that are available from the MRC IEU Open GWAS database[62,63]. For psoriasis no GWAS summary statistics was publicly available, also not in the MRC IEU Open GWAS database. In contrast, for age-related macular degeneration, we could use publicly available summary statistics from the GWAS where the association at the corresponding locus was originally reported[27]. We then matched the GWAS and eQTL SNPs by rsIDs and used all SNPs 10 kb up- and

downstream of the GWAS hit to detect colocalization using coloc. MAF estimates were used from 1000 genomes as provided from the IEU Open GWAS database.

## Statistics

Timepoint- or pathogen-specific reQTLs were determined by computing Bonferroni corrected $z$ test statistics. Further tests between two separate groups included two-tailed $t$ test or Fisher-test statistics using a $p$ value cutoff of 0.05 as indicated in the respective sections. Further details on test procedures are described in the RNAseq and gene expression analysis, eQTL, and reQTL analysis sections and in the main text.

## Reporting summary

Further information on research design is available in the Nature Portfolio Reporting Summary linked to this article.

## Data availability

Transcriptome and genotype data were collected in this study. All other data are available in the article and its Supplementary files or from the corresponding author upon request. RNA-sequencing data are accessible through GEO accession number GSE177040. Data on GWAS was collected from NHGRI-EBI GWAS Catalog (https://www.ebi.ac.uk/gwas/). Source data are provided with this paper.

## Code availability

Custom codes to generate findings in this manuscript are available at Zenodo (https://doi.org/10.5281/zenodo.7848223).

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

## Acknowledgements

We thank Cindy Reichmann for technical assistance, Jessica Noll, and Ines Leonhardt for scientific discussions throughout the project. André Heimbach, NGS Core Facility, Life & Brain Center, Bonn is acknowledged for RNA-sequencing. A.H. was supported by the German Federal Ministry of Education and Research (BMBF), program InfectControl 2020 (TFP-TV1-AS2, 03ZZ0802A) and was a member of the Jena School for Microbial Communication (JSMC). This work was supported by the Deutsche Forschungsgemeinschaft (Collaborative Research Center/Transregio 124 FungiNet - Pathogenic fungi and their human host: Networks of interaction; project C3 to O.K., A2 to J.L., INF to G.P. and S.S., and C2 to S.V. (DFG project number: 210879364) and Collaborative Research Center 1583 DECIDE - Decisions in Infectious Diseases; project C3 to O.K. (DFG project number: 492620490)) and the European Union H2020 Health (HEALTH), grant No. 847507 (to O.K.). Work in the lab of S.V. was supported by the German Ministry for Education and Science in the program Unternehmen Region (BMBF, 03Z22JN11).

## Author contributions

Conceptualization: O.K., Jo.S. Methodology: A.H., K.H., M.W., S.S. Formal analysis: S.S., V.S., J.G., T.W., Ju.S., Jo.S., J.L., O.K. Investigation: A.H., B.B., T.H., N.T. Resources: O.K., Jo.S., G.P., S.V. Writing–original draft, A.H., S.S., N.T., O.K., Jo.S. Writing–review & editing: all co-authors. Visualization: A.H., S.S., V.S., N.T. Supervision: O.K., Jo.S., G.P. Project administration: O.K., Jo.S. Funding acquisition: O.K., Jo.S., G.P., S.V.

## Funding

## Competing interests

The authors declare no competing interests.
