## [Peer Review File · Nature Communications]

Pathogen-specific innate immune response patterns are distinctly affected by genetic diversityREVIEWER COMMENTS

Reviewer #1 (Remarks to the Author):

This study describes the transcriptomes of human monocytes stimulated with fungal, Gram negative or Gram-positive bacterial pathogens. By mapping expression quantitative trait loci (eQTLs), it shows that monocytes showed a conserved response to bacterial pathogens and a distinct antifungal response. More frequently upregulated reQTL-regulated genes were NOD-like, C-type lectin, Toll-like and complement receptor-signaling pathways. reQTLs functionally characterize risk variants identified through genome-wide association studies for autoimmunity, inflammatory or infectious diseases and cancer.

This study provides novel and robust information that may help explain interindividual variation in immune response to pathogens.

Comments

Results show that, despite their distinct cell wall composition, the bacterial pathogens induced similar responses. Is that because inactivated and not live bacteria were used? I would imagine that the response substantially differ between live vs dead pathogens and this is also true for the fungus whose morphological transition is immunologically well perceived. Although the authors have clearly explained the reasons why they resorted to inactivated pathogens, still this may limit the full appreciation of the monocyte response, particularly if metabolic changes are in place. I am wondering whether the shift to aerobic glycolysis seen with inactivated *Aspergillus* hyphae is also observed with the live infecting fungal conidia and why it is not observed in response to bacteria whose macrophage response is known a reprogramming to aerobic glycolysis via HIF-1. It is also somewhat surprising that the inflammatory core immune gene response is observed in response to bacteria and less to the fungus that is also known to activate these cellular pathways in infection.

While the subsequent analysis on the causal association of reQTLs and risk variants are solid and informative, the authors may wish to better accommodate their findings in light of the above consideration before translational implications.

Reviewer #2 (Remarks to the Author):

Häder et al present a well designed experiment investigating pathogen specific gene-expression response in monocytes. They demonstrated the response to a fungal pathogen (*A. fumigatus*) was distinct to both gram-negative (*N. meningitidis*) and Gram-positive (*S. aureus*) bacteria, which showed similar responses. In addition, differences in eQTL across exposure were investigated - termed response QTL (reQTL).

It is unclear why four statistical tools are used for investigating differential expression. This is limited by the worst of the tools and can result in missing differences not found by one tool. What was the overlap between the significant genes identified by the four tools?

The eQTL analysis accounts for multiple testing, although it is unclear whether this is purely within exposure class (i.e. only correcting for number of cis regions tested for eQTL within each condition, and not the number conditions tested).

"Over all eight conditions we found 25,310 cis eQTLs where 17,851 cis eQTLs were distinct SNP-gene pairs." Distinct SNP-gene pairs are not particularly useful given we know the most associated SNP is unlikely to be causal. How many genes were associated? Is there evidence of multiple independent associations across conditions for a single gene?

The reQTL test is using the z-test as described in Kim-Hellmuth et al. However, this test assumes the

samples are independent, and is not appropriate when the same individuals are used for both conditions. The correct way to analyze the data requires fitting both conditions in a single analysis and modeling the repeated measure from each individual. Or roughly equivalently, working with the difference of gene-expression across conditions.

As a robustness analysis, it would be useful to test for reQTL between the two time-points of the non-exposed samples.

The overlap between reQTL and relevant disease GWAS hits is interesting. For the few examples presented, it would be informative to go beyond looking at LD between the most associated SNP and perform a formal colocalisation analysis, providing evidence that these loci are shared.

Reviewer #3 (Remarks to the Author):

The manuscript "Pathogen-specific innate immune response patterns are distinctly affected by genetic diversity" by Häder and Schäuble et al. describes the specific effects of three pathogens on human monocytes by means of studying response eQTLs. The authors show that inactivated *S. aureus* and *N. meningitidis* induce a shared transcriptional response in monocytes, which is divergent from the response following exposure to *A. fumigatus*.

The manuscript is well written in a punctual style when it comes to the presentation of the data and the description of the methods. Given the cells studied, monocytes, it is of no surprise that the authors identify several immune-related pathways. The collection of data is presented in a descriptive and somewhat shallow manner. The study falls short in original complexity, and in identifying the actual meaning of the findings in a pathophysiological context.

The authors claim that the study of complex stimuli is necessary to fully understand genetic regulation of transcriptional adaptation. From that perspective, the authors' choice to use inactivated pathogens is surprising. For instance, the impact of major secreted virulence factors like staphylococcal toxins is not considered in the experimental design. The claimed complexity of the current study is therefore limited, as is its novelty.

From the perspective of the host, the study design displays limited complexity as well since monocytes only are exposed to the inactivated pathogens. Thus, the impact of leukocyte interactions is not addressed. Clinical data indicate leukocytic subsets other than monocytes are at least equally if not more important during infection, for instance in neutropenic patients who are susceptible to invasive aspergillosis and recurrent staphylococcal infections.

While the need for a certain level of standardization is well taken, the authors have selected donors who fulfill very narrow inclusion criteria (male, age between 18 to 40 years). With this level of standardization, it is even more difficult to accept the authors' claim that the study reflects much complexity. On a societal note, the choice to exclusively study male donors goes against the timeframe and global public debate.

The authors stress a functional character of their approach, but the discussion highlights a major weakness of the study: for none of the findings, the actual pathophysiological mechanism is interrogated. This leaves the reader with a descriptive study and a list of 'candidates' for 'functional consequences' with 'potential impact'. The discussion contains significant overlap with the results and would benefit from abbreviating and restructuring.

Minor points:

Where does the N=215 donors come from? Is this number supported by power calculations?

The introduction needs to be abbreviated with respect to the examples of PTX3, CFH, and CFH3 since these are otherwise not addressed in the results.

Page 5, line 120: Which data support an *A. fumigatus*-induced immune response that is 'opposed' to the investigated bacteria? The data presented indicate a divergent, but not necessarily an opposed response.

Figure 3c: The third columns have dropped off the image.

Supplementary Figures 4-7: These figures, and the color codings in particular, are not intelligible and need to be revised.

Page 11, line 305: The line on diagnosis and treatment almost reads like a platitude. The authors will be aware that infections caused by *A. fumigatus*, *N. meningitidis*, and *S. aureus* have different clinical phenotypes that warrant specific treatments, irrespective of the transcriptional analyses performed in the laboratory. Obviously, this point goes even beyond the intrinsic differences in susceptibility to antimicrobial agents of the pathogens.

Pathomechanisms of diseases: I have the impression this wording is not what the authors aim to express. A pathomechanism of disease could be read as the disruption of a mechanism of disease, or, in other words: the not-disease state.

Response to reviewers' comments

Reviewer #1

“This study describes the transcriptomes of human monocytes stimulated with fungal, Gram negative or Gram-positive bacterial pathogens. By mapping expression quantitative trait loci (eQTLs), it shows that monocytes showed a conserved response to bacterial pathogens and a distinct antifungal response. More frequently upregulated reQTL-regulated genes were NOD-like, C-type lectin, Toll-like and complement receptor-signaling pathways. reQTLs functionally characterize risk variants identified through genome-wide association studies for autoimmunity, inflammatory or infectious diseases and cancer.

This study provides novel and robust information that may help explain interindividual variation in immune response to pathogens.

Comments

Results show that, despite their distinct cell wall composition, the bacterial pathogens induced similar responses. Is that because inactivated and not live bacteria were used? I would imagine that the response substantially differ between live vs dead pathogens and this is also true for the fungus whose morphological transition is immunologically well perceived. Although the authors have clearly explained the reasons why they resorted to inactivated pathogens, still this may limit the full appreciation of the monocyte response, particularly if metabolic changes are in place. I am wondering whether the shift to aerobic glycolysis seen with inactivated *Aspergillus* hyphae is also observed with the live infecting fungal conidia and why it is not observed in response to bacteria whose macrophage response is known a reprogramming to aerobic glycolysis via HIF-1. It is also somewhat surprising that the inflammatory core immune gene response is observed in response to bacteria and less to the fungus that is also known to activate these cellular pathways in infection.

While the subsequent analysis on the causal association of reQTLs and risk variants are solid and informative, the authors may wish to better accommodate their findings in light of the above consideration before translational implications.”

(i) Regarding inactivated *versus* viable pathogens as stimuli

During infection, human immune cells will be exposed to both viable and replicating and inactivated/killed pathogens. For our study, the use of inactivated pathogens is technically required (as in all comparable experimental settings published) to enable a comparable stimulation of monocytes from all 215 individuals. We have good indications that transcriptomic adaptation in response to inactivated *versus* viable pathogens is overall similar: Analyses of transcriptional regulation in human immune cells using an *ex vivo* human whole-blood assay previously found that the immune response during infection with viable microorganisms was clearly different between fungal and bacterial pathogens (Dix *et al.* (2015), PMID 25814982), similar to our current study. We rephrased large parts of the discussion and address this specific point on page 13, lines 343-347.

While we feel that use of inactivated pathogens is both technically unavoidable and biologically meaningful, we wish to further stress this as a potential limitation and have thus included the following text in the discussion of the manuscript (pages 14-15, lines 381-389):

Added text: “Another potential limitation of our study is the use of inactivated pathogens. We decided for this experimental approach to enhance comparability between the stimulations and avoid cell death during the co-incubation. Importantly, inactive pathogens will occur during natural infection and contribute to immune activation. Of note, using inactive pathogens provided a major increase in complexity compared to previous studies that assessed the effect of defined receptor-specific stimuli (Fairfax *et al.* (2014), PMID 24604202; Kim-Hellmuth *et al.* (2017), PMID 28814792). A comparison of the genetic regulation of monocyte genes in response to *N. meningitidis* (this study) versus a receptor-specific stimulus (*E. coli* LPS, Kim-Hellmuth *et al.*, 2017) showed an overlap of less than 10%, although both stimuli induced a similar proportion of reQTLs (12% after *N. meningitidis*, 17% after LPS) for all eQTLs.”

(ii) Regarding specific pathways in the transcriptomic response

Indeed, other studies analyzed aerobic glycolysis in various immune cells after LPS or bacterial stimulation (e.g., Cheng *et al.* (2016), PMID 26950237; Lachmandas *et al.* (2016), PMID 27991883; Tannahill *et al.* (2013), PMID 23535595), whereas our data do not reveal a significant effect for bacterial stimulation in a cohort of >200 different donors. Importantly, when comparing our transcriptome results to other studies, it has to be taken into account that we analyzed a far higher number of individuals than available studies addressing transcriptomic responses. As we analyzed the gene expression of 215 individuals and used $q < 0.01$ and a $\log_2FC \geq 1$ to define differentially expressed genes, our results are very robust with regard to inter-individual variation. Nevertheless, in Fig. 1d, which shows the gene expression \log_2FC of genes encoding glycolytic enzymes, HK2, PFKP and DLD are also upregulated in response to both bacteria. Within the tricarboxylic acid (TCA) cycle, which is downregulated during aerobic glycolysis, IDH1 and IDH2 are downregulated in response to all three pathogens, indicating that glycolysis and TCA cycle are also affected after bacterial stimulation. However, significant enrichment of the glycolytic pathway was only identified after exposure to *A. fumigatus*. We further agree with the reviewer that the morphology of *A. fumigatus* has an important influence on the immune response. As expected by the reviewer, Gonçalves *et al.*, 2020 (PMID 32385235) shows that monocyte-derived macrophages also trigger a shift in the direction of aerobic glycolysis after stimulation with viable *A. fumigatus*, indicating that the observed effect is independent on fungal viability. This is now specified in the manuscript on page 6, lines 118-123.

Reviewer #2

Häder et al present a well designed experiment investigating pathogen specific gene-expression response in monocytes. They demonstrated the response to a fungal pathogen (*A. fumigatus*) was distinct to both gram-negative (*N. meningitidis*) and Gram-positive (*S. aureus*) bacteria, which showed similar responses. In addition, differences in eQTL across exposure were investigated - termed response QTL (reQTL).

- 1. It is unclear why four statistical tools are used for investigating differential expression. This is limited by the worst of the tools and can result in missing differences not found by one tool. What was the overlap between the significant genes identified by the four tools?**

In the revised version of the manuscript, we have made clear that the combined use of several statistical tools is currently state of the art for RNA-seq based consensus determination of DEGs (see for example Schurch *et al.* (2016), PMID 27022035). While - as correctly pointed out by Rev. #2 - this limits the number of DEGs to a certain degree, this affects at most 8.7% of all our identified DEGs per condition. Thus, the vast majority of DEGs in our study was identified by all tools. Confirmation of data interpretation by using different DEG tools supports higher reliability in the identified DEGs (*cf.* Moulos *et al.* (2015), PMID 25452340), reporting the intersection of all tools is another layer of securing that the DEGs we report and analyzed in detail are most trustworthy. However, to accommodate the concern of Rev. #2, we included an additional supplementary figure (**Supplementary Figure 10**) showing Venn diagrams and supplementary data (**Supplementary Data 8**) to show all DEGs that are identified by at least one tool to enable further analysis by the community with any combination of DEG tools.

- 2. The eQTL analysis accounts for multiple testing, although it is unclear whether this is purely within exposure class (i.e. only correcting for number of cis regions tested for eQTL within each condition, and not the number conditions tested).**

We thank the reviewer for this comment. Indeed, in the previous version of our manuscript it was not sufficiently clear how we have corrected for multiple testing. Our correction procedure was two-staged and followed an approach described in Kim-Hellmuth *et al.*, 2017 (PMID 28814792). First, we performed a local correction within each condition and applied a permutation of the lead *cis* eQTL for each eGene (one eQTL per gene within each condition). Again within each condition, we then performed a global correction considering all genes tested and applied a false discovery rate (FDR) procedure. In the revised version of the manuscript, we now describe the multiple testing procedure in more detail in the section *Methods* and subsection *eQTL analysis* (page 21, lines 558-563):

Original text: "Identification of eGenes (one eQTL per gene within each condition) was conducted using the permutation pass of QTLTools with the setting "—permute 1000." Permutation p-values were corrected for multiple testing applying a false discovery rate of 5%."

New version: “Identification of eGenes (one eQTL per gene within each condition) was conducted according to Kim-Hellmuth *et al.* (2017, PMID 28814792) using the permutation pass of QTLTools with the setting “—permute 1000.” Permutation p-values were gained by a beta approximation to correct for testing multiple SNPs as *cis* eQTL at each locus (local correction). Again within each condition, we then applied the false discovery rate (FDR) procedure by Storey and Tibshirani (2003, PMID 12883005) using the R-package qvalue to correct genome-wide for testing multiple genes (global correction).”

- 3. "Over all eight conditions we found 25,310 *cis* eQTLs where 17,851 *cis* eQTLs were distinct SNP-gene pairs." Distinct SNP-gene pairs are not particularly useful given we know the most associated SNP is unlikely to be causal. How many genes were associated? Is there evidence of multiple independent associations across conditions for a single gene?**

We also thank the reviewer for this comment, which we feel has helped to significantly improve our manuscript. We now specify the number of genes for which we found eQTL effects. These are 6,865 eGenes. Accordingly, we have added the following sentence in the section *Methods* and subsection *eQTL analysis* (page 21, lines 565-566):

Added text: “This refers to 6,865 different eGenes and shows that the majority of *cis* eQTLs are present across different conditions.”

In addition, as suggested by this reviewer, we tested to what extent genes that are regulated across pathogens have different *cis* reQTLs. We restricted this analysis to reQTLs, because they are the main focus of the present study. For this, we defined reQTLs for eGenes as independent when the effects are caused by different SNPs that show only low LD to each other ($r^2 < 0.2$). The findings are presented in the newly added **Supplementary Data 4** and revealed (i) that the majority of eGenes are not independently regulated across conditions (between 90.70% and 96.16%) and (ii) that different reQTLs are not more frequently present when comparing *A. fumigatus* and bacteria as stimuli. In addition to **Supplementary Data 4**, we have added the following sentence in the section *Results* and subsection *reQTL-regulated expression differs for fungal and bacterial stimulation* (page 7, lines 169-171):

Added text: “However, when genes were significantly regulated both upon *A. fumigatus* and bacteria stimulation the same – and not independent – reQTLs seem to be active at the majority of loci (between 90.70% and 96.16%) (**Supplementary Data 4**).”

- 4. The reQTL test is using the z-test as described in Kim-Hellmuth et al. However, this test assumes the samples are independent, and is not appropriate when the same individuals are used for both conditions. The correct way to analyze the data requires fitting both conditions in a single analysis and modeling the repeated measure from each individual. Or roughly equivalently, working with the difference of gene-expression across conditions.**

The reviewer raises an important point that we have not adequately addressed in the previous version of our manuscript by only referring to our work by Kim-Hellmuth *et al.*, 2017 (PMID 28814792). In the latter study, we also used the z-test (beta comparison baseline vs. stimulus) to identify reQTLs. However, in this study we also compared the results of this approach (reQTL) with those obtained when using differential expression upon stimulation as a quantitative trait (diffQTL). The result of this comparison is shown as Supplementary Fig. 5b in Kim-Hellmuth *et al.*, 2017 (PMID 28814792). In this study, we used three different stimuli (LPS, RNA, MDP) and two different time points (90min, 6h). On the x-axis of the scatter plots eQTL p-values are shown that were calculated using z-test or beta comparison (reQTL) and on the y-axis eQTL p-values are shown that were calculated using differential expression upon stimulation as phenotype (diffQTL). The Spearman's rank correlation coefficient is shown in the upper right corner of each plot and indicates a nearly perfect correlation between both methods.

In the revised version of the manuscript we are now addressing this point in the section *Methods* and subsection *reQTL analysis* (page 21, lines 574-579):

Added text: "Of note, previous studies (e.g., Kim *et al.* (2014), PMID 25327457) have used differential expression upon stimulation as quantitative trait for the detection of reQTLs (called diffQTL). However, in our study by Kim-Hellmuth *et al.*, 2017 (PMID 28814792) we have also calculated diffQTLs for all reQTLs identified by β -comparison and used Spearman correlation as a measure of similarity. This showed that both methods lead to nearly identical results."

5. As a robustness analysis, it would be useful to test for reQTL between the two time-points of the non-exposed samples.

We thank the reviewer for this suggestion, which further illustrates the robustness of our reQTLs. Out of 3,279 transcripts that were present in 3h and 6h unstimulated monocytes, only six (0.2%) showed significant results comparing the regression coefficients at both time points in the z-test. This refers to the transcripts *SKIV2L2*, *TOMM22*, *JMJD6*, *RPS15AP34*, *TNIP2* and *PPP2R2A*. Of them, only 2 showed reQTL effects in our analysis. In the revised version of the manuscript we are now addressing this point in the section *Methods* and subsection *reQTL analysis* (page 22, lines 589-593):

Added text: "Finally, we tested the robustness of the stimulus-dependent reQTLs. For this, we used all 3,279 transcripts that are present in 3h and 6h unstimulated monocytes. Of them, only 6 transcripts (0.2%) showed significant results when we compared the regression coefficients at both time points with the z-test. The data showed that the reQTLs in our study were almost entirely due to the stimulations and were barely artificially induced."

6. The overlap between reQTL and relevant disease GWAS hits is interesting. For the few examples presented, it would be informative to go beyond looking at LD between the most associated SNP and perform a formal colocalisation analysis, providing evidence that these loci are shared.

We agree with the reviewer that a colocalisation analysis represents an alternative approach to determine whether identified reQTLs constitute risk variants for multifactorial diseases. For this, we have focused on the reQTLs that are presented in the section *Results*, because it is difficult to access GWAS data for all 60 overlapping GWAS-reQTL pairs that have been identified. As recommended by the reviewer, we used coloc (Giambartolomei *et al.* (2014), PMID 24830394) as method as well as GWAS data for multiple sclerosis, inflammatory bowel disease, systemic lupus erythematosus and primary biliary cholangitis, which are publicly available from the MRC IEU Open GWAS database (Lyon *et al.* (2021), PMID 33441155). However, for psoriasis no GWAS summary statistics was publicly available, which showed association for the corresponding locus, also not in the MRC IEU Open GWAS database. Only for age-related macular degeneration we could use the summary statistics from the GWAS where the association at the corresponding locus was originally reported (Fritsche *et al.* (2013), PMID 23455636), because data were publicly available. In the coloc analysis we found evidence for colocalization for *IRF8* in monocytes upon 6h exposure with *N. meningitidis* or *S. aureus* with multiple sclerosis ($H_{4,NM} = 0.997$, $H_{4,SA} = 0.997$) and systemic lupus erythematosus ($H_{4,NM} = 0.989$, $H_{4,SA} = 0.990$), but only moderate evidence for inflammatory bowel disease ($H_{4,NM} = 0.577$, $H_{4,SA} = 0.616$). In contrast, under these conditions we found no evidence for colocalization with primary biliary cholangitis ($H_{4,NM} = 0.140$, $H_{4,SA} = 0.004$). However, this estimate is not based on the GWAS that reported this association (Mells *et al.* (2011), PMID 21399635), because corresponding data were not publicly available (see above). For age-related macular degeneration we found strong colocalization for *TNFRSF10A* in monocytes upon 6h exposure with *N. meningitidis* or *S. aureus* ($H_{4,NM} > 0.999$, $H_{4,SA} > 0.999$). Unfortunately, we could not test for colocalization for *NIPAL4* and psoriasis, because no GWAS summary statistic was publicly available for this trait (see above). In the revised version of the manuscript, we are now presenting the colocalization findings as **Supplementary Data 7** as well as in the section *Results* and subsection *reQTL analysis reveals potential pathomechanisms leading to noncommunicable diseases* (page 12, lines 310-321):

Added text: “Finally, we performed a colocalization analysis using coloc for all above-mentioned variants (**Supplementary Data 7**). We found evidence for colocalization for *IRF8* in monocytes after 6h exposure to *N. meningitidis* and *S. aureus* with multiple sclerosis ($H_{4,NM} = 0.997$, $H_{4,SA} = 0.997$) and systemic lupus erythematosus ($H_{4,NM} = 0.989$, $H_{4,SA} = 0.990$), but only moderate evidence with inflammatory bowel disease ($H_{4,NM} = 0.577$, $H_{4,SA} = 0.616$). In contrast, under these conditions we found no evidence for colocalization with primary biliary cholangitis ($H_{4,NM} = 0.140$, $H_{4,SA} = 0.004$). However, this estimate is not based on the GWAS that reported this association (Mells *et al.* (2011), PMID 21399635), because corresponding data were not publicly available. For age-related macular degeneration we found strong colocalization for *TNFRSF10A* in monocytes following 6h exposure to *N. meningitidis* or *S. aureus* ($H_{4,NM} > 0.999$, $H_{4,SA} > 0.999$). Unfortunately, we could not test for

colocalization for *NIPAL4* and psoriasis, because no GWAS summary statistic was publicly available for this trait, which showed association for the corresponding locus.“

In addition, we have added a paragraph describing the coloc method in the section *Methods* and subsection *Overlap between reQTLs and GWAS catalog* (page 23, lines 627-637):

Added text: “Finally, we performed a colocalization analysis using coloc v5.1.1 (Giambartolomei *et al.* (2014), PMID 24830394) for the GWAS-reQTL pairs that are presented in the result section. For this, we used the summary statistics for the corresponding diseases – with the exception of psoriasis and age-related macular degeneration – that are available from the MRC IEU Open GWAS database (Lyon *et al.* (2021), PMID 33441155; Elsworth *et al.* (2020), <https://doi.org/10.1101/2020.08.10.244293>). For psoriasis no GWAS summary statistics was publicly available, also not in the MRC IEU Open GWAS database. In contrast, for age-related macular degeneration we could use publicly available summary statistics from the GWAS where the association at the corresponding locus was originally reported (Fritsche *et al.* (2013), PMID 23455636). We then matched the GWAS and eQTL SNPs by rsIDs and used all SNPs 10 kb up- and downstream of the GWAS hit to detect colocalization using coloc. MAF estimates were used from 1000 genomes as provided from the IEU Open GWAS database.”

Reviewer #3

The manuscript “Pathogen-specific innate immune response patterns are distinctly affected by genetic diversity” by Häder and Schäuble et al. describes the specific effects of three pathogens on human monocytes by means of studying response eQTLs. The authors show that inactivated *S. aureus* and *N. meningitidis* induce a shared transcriptional response in monocytes, which is divergent from the response following exposure to *A. fumigatus*.

- 1. The manuscript is well written in a punctual style when it comes to the presentation of the data and the description of the methods. Given the cells studied, monocytes, it is of no surprise that the authors identify several immune-related pathways. The collection of data is presented in a descriptive and somewhat shallow manner. The study falls short in original complexity, and in identifying the actual meaning of the findings in a pathophysiological context.**

eQTL studies are a powerful tool to analyze the impact of genetic variation across humans on the gene expression, that enables the identification of cellular mechanisms that underlie human genetic diseases. With that aim, several eQTL studies analyzed various normal and non-diseased human tissues (e. g. GTEx Consortium *et al.* (2020), PMID 32913098; Schmiedel *et al.* (2018), PMID 30449622). Compared to eQTL-studies, that determine the influence of genetic variation in an unstimulated context, the stimulation with whole microorganisms increases the complexity of the analysis and allows the identification of genetic regulatory effects that change in response to a stimulation. We agree with Rev. #3 that our setup cannot reflect all facets of the *in vivo* complexity. However, we are convinced that our approach is an important step forward towards our understanding of discriminatory immunity in response to different types of pathogens - this together with the identification of new functional links between genetic variants and gene expression under infection-mimicking conditions is the novelty of our study. To better describe the novelties of our study as well as to discuss its implications we substantially rewrote large parts of our manuscript to address all raised points by this and all other reviewers.

- 2. The authors claim that the study of complex stimuli is necessary to fully understand genetic regulation of transcriptional adaptation. From that perspective, the authors’ choice to use inactivated pathogens is surprising. For instance, the impact of major secreted virulence factors like staphylococcal toxins is not considered in the experimental design. The claimed complexity of the current study is therefore limited, as is its novelty.**

The use of inactivated pathogens is currently common standard in all comparable experimental settings. Isolated monocytes are unable to prevent fungal filamentation after phagocytosis, which leads to immune cell lysis and fungal escape (Loeffler *et al.* (2009), PMID 19074652), and rapid bacterial replication during primary confrontation. Therefore, the use of inactivated pathogens is necessary to avoid cell death and analyze a bona fide transcriptional response

of host immune cells. For reference, we would like to point towards a recent study published in *Nature Communications* by Oelen *et al.* (2022, PMID 35672358) that also used inactivated pathogens to investigate gene expression regulation in stimulated PBMCs.

Importantly, during infection the human immune system is unavoidably exposed to both viable and dead microorganisms. Furthermore, there is good evidence that despite some differences, the response towards inactivated *versus* viable pathogens is highly comparable (see Rev. #1). Thus, while we are convinced that our study is unique in determining system-wide pathogen-specific differences using a systematic analysis of genetic regulation in immune activation, we have discussed this potential limitation of our study in the rephrased discussion of our manuscript (*cf.* response to Rev. #1, pages 14-15, lines 381-389).

3. From the perspective of the host, the study design displays limited complexity as well since monocytes only are exposed to the inactivated pathogens. Thus, the impact of leukocyte interactions is not addressed. Clinical data indicate leukocytic subsets other than monocytes are at least equally if not more important during infection, for instance in neutropenic patients who are susceptible to invasive aspergillosis and recurrent staphylococcal infections.

While we agree with the reviewer, that other immune cell types are highly relevant in the defense against bacterial / fungal infection, we strongly argue that the use of purified monocytes is both experimentally relevant and biologically meaningful:

(i) The use of mixed cell populations is in our view not a feasible approach: Variability due to fluctuating proportions of different immune cell populations in PBMCs or whole blood would directly interfere with data analysis. Importantly, eQTL effects differ strongly depending on the cell type (Fairfax *et al.* (2012), PMID 22446964; GTEx-Consortium *et al.* (2020), PMID 32913098). It is well documented that activity of eQTLs is often cell-type-specific (van der Wijst *et al.* (2018), PMID 29610479), which prevents analyses required for our study in a non-purified cell population.

(ii) The choice for monocytes was made as these cells are the most transcriptionally active population in peripheral blood. Pathogen-activated monocytes are of central importance in the pathophysiological processes of systemic infection and sepsis, as they can phagocytose microorganisms, process them intracellularly, and initiate an adaptive immune response as professional antigen-presenting cells. Upon stimulation, monocytes and its derivatives exhibit a high degree of transcriptional plasticity, a fundamental characteristic that is, in this extent, not present in neutrophils (Auffray *et al.* (2009), PMID 19132917; Serbina *et al.* (2008), PMID 18303997).

(iii) In addition to their importance in the immune response after infections, monocytes also play a central role in non-infectious inflammatory processes, such as arteriosclerosis, Alzheimer's disease and multiple sclerosis (Karlmark *et al.* (2012), PMID 24672677). In our study, we were able to identify reQTLs that are in high linkage disequilibrium ($LD > 0.8$) to risk

SNPs of GWAS for these diseases. Furthermore, other than human neutrophils, which have a limited repertoire of transcriptional activation, human monocytes show a very high plasticity of transcriptional activity that is directly linked to their functional impact. Thus, we feel that monocytes are an appropriate cell type for the initial analysis of eQTLs under infection conditions.

With the increasing development of single-cell RNA-sequencing, the method could become more cost-effective and thus be suitable for future eQTL studies in mixed cell populations. This technique was used by Oelen *et al.* (2022, PMID 35672358) to investigate genetic regulation of gene expression in stimulated PBMCs on the single cell level. Although Oelen *et al.* and our study focus on different questions - Oelen *et al.* aimed at differences of the transcriptional response between cell types, whereas our study targeted pathogen-dependent effects within one cell type - the achieved results of both studies are complementary. Comparable to our study, Oelen *et al.* used three different pathogens, *Candida albicans*, *Mycobacterium tuberculosis* and *Pseudomonas aeruginosa*, in an inactivated state (heat-killed). We performed a thorough comparison of both studies. Key findings are:

- Gene expression responses after pathogen stimulation as analyzed by Oelen *et al.* revealed a strong cell-type-specificity. Myeloid cells (monocytes and DCs) had the highest and most unique number of differentially expressed genes (DEGs) upon stimulation - a result that provides further evidence that choosing monocytes as a cell type for response eQTL (reQTL) analyses is a valid and relevant approach.
- Consistent in both studies, the number of DEGs in monocytes increased from early (3h, both studies) to late time point (24h, Oelen *et al.*; 6h, our study) of pathogen stimulation.
- Despite our conservative bioinformatics analysis (see also Rev. #2, Point #1) our study revealed a substantial higher number of DEGs than the study by Oelen *et al.* Most likely, this is due to the larger number of participants in our study (215 individuals *versus* 120 individuals in the study by Oelen *et al.*) and the resulting higher statistical power.
- We performed an additional functional enrichment analysis of the Oelen *et al.* data using the same workflow as in our study to maximize comparability and to compute enriched KEGG pathways that were not reported in Oelen *et al.* Using DEGs upon stimulation we found **all pathways** that have been identified in our study (3h) and belong to the pathogen-independent core immune response also in monocytes (3h) from the Oelen *et al.* study.
- Whereas our DEGs upon stimulation discriminate fungal and bacterial pathogens (see **Fig. 1a** and **1b**), DEGs in the Oelen *et al.* study were very similar among all three stimulations (3h, monocytes). This is probably also due to the greater statistical power of our study or the fact that Oelen *et al.* was more interested in the cell type-specific immune response, whereas we are more interested in the pathogen-specific immune response. Thus, both datasets are highly complementary.

- Despite methodological differences and the use of different pathogens in both studies, the majority of reQTL-regulated genes in the monocyte subset from Oelen *et al.* overlap with our data (504/617 *cis* reQTLs, see new **Supplementary Fig. 3**). The fact that we could identify the vast majority of all reQTLs identified by Oelen *et al.* also indicates a low impact of sex differences in the genetic regulation of the transcriptional response since Oelen *et al.* isolated PBMCs from whole blood of males and females. In addition to overlapping reQTLs, we were able to identify a large number of additional reQTLs (in total 3,929 *cis* reQTLs), which is due to the greater statistical power of our study.

We refer to the study by Oelen *et al.* at various points in the current version of the manuscript:

Introduction: page 3, lines 60-62

Added text: "... and are often cell type-specific. With regard to peripheral blood immune cells, reQTLs in monocytes have been shown to outnumber those in other peripheral blood mononuclear cells (Oelen *et al.* (2022), PMID 35672358)."

Results, subsection *reQTLs modulate expression of 11% of regulated genes*: page 6, lines 141-145

Added text: "A comparison with recent data by Oelen *et al.* showed more than 80% overlap of their eQTL-regulated monocyte genes (504/617 *cis* eQTLs, 3h stimulated monocytes) with our data (**Supplementary Fig. 3**), despite the use of different pathogens (Oelen *et al.* (2022), PMID 35672358). These correspond to only a fraction of all *cis* eQTLs detected in this study, which had 3,929 additional eQTL-regulated genes after 3h pathogen exposure."

Discussion: page 13, lines 335-338

Added text: "The use of purified immune cell populations is supported by the strong cell-type-specificity of the gene expression response and its genetic regulation in PBMCs upon pathogen stimulation. Among all peripheral blood immune cells, monocytes are the most promising target due to a higher number of DEGs and reQTLs compared to other cell types (Oelen *et al.* (2022), PMID 35672358)."

Due to technical limitations, single-cell RNA-sequencing currently addresses the difference between host cell types, rather than trigger-dependent differences. While this is clearly a valuable approach, it is currently not suited for our study aims. Importantly, despite the technical differences, Oelen *et al.* clearly show that reQTLs in monocytes outnumber those in other peripheral blood cells, supporting our focus on this cell type.

4. While the need for a certain level of standardization is well taken, the authors have selected donors who fulfill very narrow inclusion criteria (male, age between 18 to 40 years). With this level of standardization, it is even more difficult to accept the authors' claim that the study reflects much complexity. On a societal note, the choice to exclusively study male donors goes against the timeframe and global public debate.

We fully agree with the reviewer that a study design with both sexes would have been preferable. However, biological sex is associated with major differences in immune cell gene expression (Schmiedel *et al.* (2018), PMID 30449622; Khramtsova *et al.* (2019), PMID 30581192; Moore *et al.* (2021), PMID 34903727). Furthermore, it is uncertain how much additional variance of the immune response is introduced by cyclical changes of sex hormones in women (Klein & Flanagan (2016), PMID 27546235). We have now explicitly added discussion of this limitation to the manuscript (page 14, lines 356-361).

In addition, we were able to experimentally confirm the activity of two identified reQTLs on a functional level for female donors, showing that at least some important reQTLs identified in this study are also active in a female background (see new **Figure 4**). For this, freshly isolated monocytes were stimulated with pathogens and either directly tested for NOD1-dependent release of IL-1 β or confronted with T cells to analyze the effect of differences in reQTL-regulated CD86 expression on T cell activation by IFN- γ release. Results from the functional *ex vivo* assays show that cytokine secretion upon pathogen stimulation is genotype-dependent and thereby validate our results from the reQTL analysis. In addition, we show that these effects are also present in females and are, thus, not restricted to males. For description of the functional *ex vivo* assays we included the following parts in the revised manuscript:

Results, subsection reQTLs are functionally relevant in PRR signaling: pages 9-10, lines 228-248

Added text: "reQTLs are functionally relevant in PRR signaling"

Using *ex-vivo* assays we examined the functional effect of two identified reQTLs on cytokine production using cells from female donors. For functional analysis of the *NOD1* rs62447420 SNP we isolated monocytes from freshly drawn blood and stimulated them with *S. aureus* and *A. fumigatus* to quantify activation-dependent IL-1 β secretion. In accordance with results obtained in our reQTL study, monocytes from donors carrying the T variant of the *NOD1* rs62447420 secreted significantly more IL-1 β compared to donors carrying the C variant of this SNP in response to pattern recognition of *S. aureus* (**Fig. 4b**), whereas only a slight effect could be observed after confrontation with *A. fumigatus* (**Fig. 4c**). Stimulation with LPS served as positive control for NOD1 receptor activation and showed a strong IL-1 β release, especially in donors carrying the T variant (**Fig. 4a**). We additionally assessed functional relevance of the reQTL effect on *CD86* gene expression, that encodes a costimulatory molecule necessary for T cell activation. We isolated monocytes as well as T cells from freshly drawn blood and stimulated these with *S. aureus* and *A. fumigatus* to measure CD86-driven T cell

activation by the resulting IFN γ secretion. Carriers of the A allele showed a trend towards increased IFN γ levels in response to both pathogens, although results were not significant (**Fig. 4e** and **4f**). T cell activation was validated by stimulation with CD3 antibody (**Fig. 4d**).

Cells used for functional confirmation were isolated from female donors to additionally test the influence of the donor's sex. Although not tested for all initially identified reQTLs, our results indicate that their effects are likely not specific to one sex, but may underly a sex-independent immune response."

Discussion: page 14, lines 361-380

Added text: "Despite these reported sex-associated differences, we were able to confirm selected reQTLs for female donors, showing that at least some important reQTLs identified in this study are also active in a female background. The reQTL for *NOD1* in monocytes was identified after stimulation with *N. meningitidis* and *S. aureus*. *NOD1* encodes an intracellular PRR that drives proinflammatory and antimicrobial responses (Caruso *et al.* (2014), PMID 25526305). Functional relevance of this reQTL in PRR signaling was validated by stimulation of *NOD1* signaling in primary monocytes with *S. aureus*, showing an increased cytokine release by donors carrying the allele associated with higher *NOD1* expression. In addition, functional reQTL effect was shown for *CD86* upon *A. fumigatus* and *S. aureus* exposure. *CD86* encodes a co-signaling molecule on the surface of antigen-presenting cells that controls the T cell response to antigens in conjunction with T cell receptor signals, inducing adaptive immune responses (Chen & Flies (2013), PMID 23470321). Both *CD86* and its paralog *CD80* were upregulated in a model of experimental allergic aspergillosis, demonstrating a significant role for these co-stimulatory molecules in onset, persistence and progression of immune responses (Barrios *et al.* (2005), PMID 16232210). Interestingly, this reQTL is in linkage disequilibrium (LD, $r^2 = 0.81$) with rs75557865, a risk SNP in GWAS for allergic disease (asthma, hay fever or eczema) (Ferreira *et al.* (2017), PMID 29083406). The reQTL effect was also reported in a study on IFN γ -stimulated monocytes (Fairfax *et al.* (2014), PMID 24604202), showing the robustness of the reQTL effect on *CD86*. Thus, this reQTL is a possible candidate for functional consequences for allergic diseases. We showed that part of our findings can be confirmed with female-derived samples, which allowed us to conclude that sex-independent reQTL effects exist. Generalizing on our findings, however, warrants further equally sized study cohorts including both sexes."

In line with this, a comparison with data by Oelen *et al.* (2022, PMID 35672358), which investigated pathogen-stimulated PBMCs from whole blood of men and women, indicates a low impact of sex differences on genetic regulation of gene expression: The majority of their eQTL-regulated monocytic genes (504/617 *cis* eQTLs) were also present in our data (see new **Supplementary Figure 3**).

- 5. The authors stress a functional character of their approach, but the discussion highlights a major weakness of the study: for none of the findings, the actual pathophysiological mechanism is interrogated. This leaves the reader with a descriptive study and a list of ‘candidates’ for ‘functional consequences’ with ‘potential impact’. The discussion contains significant overlap with the results and would benefit from abbreviating and restructuring.**

We extensively rephrased the discussion of our manuscript. It now includes functional consequences for the reQTL-regulated genes *IRF8*, *NIPAL4* and *TNFRSF10*, that are described in the literature (page 15, lines 394-413). We chose these reQTL-regulated genes, because the corresponding reQTLs are identical to disease-associated SNPs, allowing the link between the risk allele and the influence on the gene expression literature:

Added text: “The strongest reQTL that was linked to an association study was in LD with a risk variant for psoriasis and influenced *NIPAL4* expression. Mutations in *NIPAL4* genes are described in patients with congenital ichthyosis, a monogenic disease with cutaneous manifestation but unknown underlying pathogenetic mechanisms (Lefèvre *et al.* (2004), PMID 15317751). Gene expression profiling of skin from patients with congenital ichthyosis identified significant upregulation of psoriasis hallmark genes, while *Nipal4*-knockout mice exhibited neonatal lethality due to skin barrier defects (Murase *et al.* (2020), PMID 31836270; Honda *et al.* (2018), PMID 29174370). A disease-associated SNP from GWAS for primary biliary cholangitis, inflammatory bowel disease, systemic lupus erythematosus and multiple sclerosis was in strong LD to a reQTL that regulated expression of *IRF8*. *IRF8*, a key factor for dendritic cell, monocyte and macrophage maturation, was described to influence the genetic risk for susceptibility to several chronic inflammatory diseases (Salem *et al.* (2020), PMID 32232558; Sichien *et al.* (2016), PMID 27637148; Hagemeyer *et al.* (2016), PMID 27412700). siRNA-mediated inhibition of *Irf8* *in vitro* and *in vivo* reduces the intensity of the inflammatory response (mediated by TNF- α , IL-6, IL-12/IL-23 and IL-1 β) and the associated pathology in a mouse model of intestinal colitis. Finally, a reQTL that was linked to a risk-SNP for age-related macular degeneration was found to regulate *TNFRSF10A* gene expression. Binding of TRAIL to the encoded death receptor TRAILR1 is known to induce apoptosis (Johnstone *et al.* (2008), PMID 18813321). TRAILR1 is present in cultured retinal pigment epithelium cells and late manifestation of age-related macular degeneration is characterized by atrophy of the retinal pigment epithelium, followed by degeneration of the choriocapillaris (McLeod *et al.* (2009), PMID 19357355). Our data showed that the risk allele for age-related macular degeneration is linked to increased *TNFRSF10A* expression compared to the opposite allele.”

Minor points:

- 6. Where does the N=215 donors come from? Is this number supported by power calculations?**

We thank the reviewer for this comment. We adapted our sample size from sample sizes used in previous eQTL studies where > 1,000 eQTLs have been identified (e. g. Kim *et al.* (2014),

PMID 25327457 analysed 137 participants and Nédélec *et al.* (2016), PMID 27768889 analysed 175 participants). In addition, we have carried out a power analysis under the assumption of different minor allele frequencies (MAF) and eQTL effect sizes that is shown in **Supplementary Figure 9**. In the revised version of the manuscript we are referring to this figure in the section *Methods* and subsection *Donor collection* (page 17, lines 444-447):

Added text: “**Supplementary Fig. 9** depicts the power analysis of our samples for the identification of eQTL effects under the assumption of different MAF and effect sizes. The power analysis was performed with powerEQTL v0.3.4 (Dong *et al.* (2021), PMID 34009297). Fixed parameters were set as follows: $\sigma_{y^2} = 0.13$, $n_{\text{Tests}} = 30,000,000$.”

7. The introduction needs to be abbreviated with respect to the examples of PTX3, CFH, and CFH3 since these are otherwise not addressed in the results.

We abbreviated the introduction and deleted the paragraph that lists the genes *PTX3*, *CFH* and *CFH3* and their relevance in *A. fumigatus* and *N. meningitidis* infection.

8. Page 5, line 120: Which data support an *A. fumigatus*-induced immune response that is ‘opposed’ to the investigated bacteria? The data presented indicate a divergent, but not necessarily an opposed response.

We thank the reviewer for this comment. This has been corrected. The full sentence in the manuscript now reads (page 5, lines 115-116):

“Compared to the bacterial pathogens, *A. fumigatus* induced a specific immune response divergent to the investigated bacteria.”

9. Figure 3c: The third columns have dropped off the image.

The third column shows *PPP3CA* expression (6h) for carriers of the alleles AA for SNP rs17030831. In our cohort only one individual had this genotype. Consequently, the third column only consists of a point instead of a violin plot.

10. Supplementary Figures 4-7: These figures, and the color codings in particular, are not intelligible and need to be revised.

We agree with the reviewer that it is difficult for readers to grasp all information from these figures. Although the overall design of the figures is based on the standard output of a commonly applied tool, we have further improved the layout. In the gene boxes, we reduced the amount of information (gene expression for *A. fumigatus*, *N. meningitidis* and *S. aureus*). With stars we highlight the occurrence of reQTLs. The color code was changed to improve the

readability. Instead of traditional green to red we adapted the HCL (hue-chroma-luminance) “Blue-Red 2” color palette, which was created with improved differentiability in mind. HCL coordinate based color palettes were shown to be superior over regular RGB based color palettes (cf. <https://arxiv.org/abs/1903.06490>). Furthermore, we increased the size of the figures by cubic interpolation using GIMP v2.10.18.

11. Page 11, line 305: The line on diagnosis and treatment almost reads like a platitude. The authors will be aware that infections caused by *A. fumigatus*, *N. meningitidis*, and *S. aureus* have different clinical phenotypes that warrant specific treatments, irrespective of the transcriptional analyses performed in the laboratory. Obviously, this point goes even beyond the intrinsic differences in susceptibility to antimicrobial agents of the pathogens.

We agree and deleted this sentence.

12. Pathomechanisms of diseases: I have the impression this wording is not what the authors aim to express. A pathomechanism of disease could be read as the disruption of a mechanism of disease, or, in other words: the not-disease state.

We agree with Rev. #3 and have changed the wording in the sentences concerned.

(i) Original text (headline): “reQTL analysis reveals potential pathomechanisms of noncommunicable diseases.”

New version (page 11, line 270, headline): “reQTL analysis reveals potential mechanisms leading to noncommunicable diseases”

(ii) Original text: “However, with only GWAS, identifying disease-causing pathomechanisms and corresponding risk genes is nearly impossible.”

New version (page 11, lines 272-273): “However, identifying causal mechanisms and corresponding risk genes is nearly impossible by only relying on GWAS.”

(iii) Original text: “The reQTLs we identified may contribute to understanding the pathomechanisms of diseases that are not directly caused by microbial infections.”

New version (page 11, lines 275-277): “The reQTLs we identified may contribute to understanding the molecular mechanisms leading to diseases that are not caused by microbial infections.”

REVIEWERS' COMMENTS

Reviewer #1 (Remarks to the Author):

The Authors have properly addressed the concerns I have raised.

Reviewer #2 (Remarks to the Author):

The authors have performed substantial additional analysis to address the concerns raise during my review. All queries have been addressed in detail, and provide satisfactory answers. I have no further issues with the manuscript.

Reviewer #3 (Remarks to the Author):

I thank the authors for their responses and have no further comments.